# L-SR1: Learned Symmetric-Rank-One Preconditioning

**Gal Lifshitz** [1]  **Shahar Zuler** [1]  **Ori Fouks** [1]  **Dan Raviv** [1]

## Abstract

End-to-end deep learning has achieved impressive results but often relies on large labeled datasets, exhibits limited generalization to unseen scenarios, and incurs substantial computational cost. Classical optimization methods, in contrast, are more data-efficient and lightweight but frequently suffer from slow convergence. Learned optimizers aim to bridge this gap, yet existing approaches have focused primarily on first-order methods, while learned second-order optimization has received much less attention. We introduce L-SR1, a learned second-order optimizer inspired by the classical Symmetric Rank-One (SR1) method. At its core, L-SR1 employs a Projection-Guided Secant Mechanism (PGSM) that generates positive semi-definite preconditioners and biases meta-training toward the quasi-Newton secant relation. Through controlled analytic benchmarks, we study stability, generalization across problem dimensions, and search direction quality, and further evaluate L-SR1 on Monocular Human Mesh Recovery (HMR), where it outperforms both classical and learned optimization-based baselines. With a compact model and no reliance on task-specific fine-tuning or annotated data, L-SR1 demonstrates strong generalization and can be integrated into a broad range of iterative optimization problems to accelerate convergence and reduce the required number of iterations.

## 1. Introduction

End-to-end deep learning has demonstrated significant power but is constrained by its reliance on large labeled datasets and limited ability to generalize to unseen scenarios. Furthermore, increased model sizes featured in recent works pose a limitation as they demand high compute and memory resources. In contrast, classical optimization excels in data-scarce settings and features a low memory stamp, but often suffers from long runtime due to its iterative nature. To address this, extensive research has focused on accelerating convergence, with optimization methods broadly categorized into first-order and second-order approaches.

First-order methods, such as Adam (Kingma & Ba, 2014) and Nesterov Accelerated Gradient (NAG) (Nesterov, 1983; Sutskever et al., 2013), rely on estimated gradient momentums for parameter updates. Second-order methods, such as Symmetric-Rank-One (SR1) (Conn et al., 1991) and Broyden-Fletcher-Goldfarb-Shanno (BFGS) (Liu & Nocedal, 1989), utilize approximations of the inverse Hessian matrix (Boyd et al., 2004; Bertsekas, 1999). While more computationally intensive, second-order methods typically achieve faster convergence by accurately capturing the underlying structure of the objective function and exploiting dependencies between variables.

Learned optimization has recently emerged as a promising field that leverages deep learning to enhance traditional optimization methods. These approaches incorporate trainable deep neural network (DNN) architectures—such as Multi-Layer Perceptrons (MLPs) (Andrychowicz et al., 2016; Li & Malik, 2016; Song et al., 2020), Recurrent Neural Networks (RNNs) (Andrychowicz et al., 2016), Transformers (Gärtner et al., 2023), and hybrid models (Metz et al., 2020)—into optimization frameworks. Once trained on specific objectives, these learned optimizers exhibit significantly faster convergence. Although learned optimization has been widely explored for first-order methods (Metz et al., 2022), several works have also considered second-order approaches (Li et al., 2020; Liao et al., 2023). Nevertheless, the integration of learnable components with second-order methods remains relatively underexplored.

By accelerating convergence, learned optimizers offer a compelling path to bridging classical optimization techniques with modern deep learning-based approaches in computer vision tasks.

Monocular Human Mesh Recovery (HMR) seeks to estimate a 3D human mesh from a single 2D image—a fundamentally ill-posed problem due to the inherent loss of depth information. Historically, in the absence of large-scale annotated datasets, optimization-based methods dominated the field (Bogo et al., 2016; Pavlakos et al., 2019).

[1]Tel Aviv University, Tel Aviv, Israel. Correspondence to: Gal Lifshitz <lifshitz@mail.tau.ac.il>.

*Proceedings of the 43rd International Conference on Machine Learning*, Seoul, South Korea. PMLR 306, 2026. Copyright 2026 by the author(s).

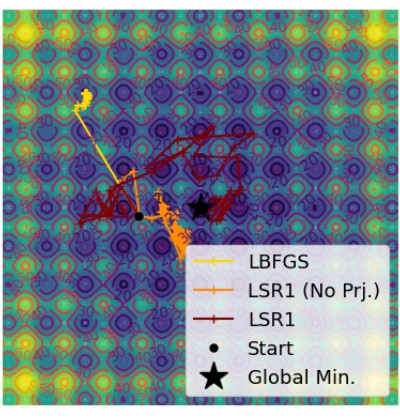
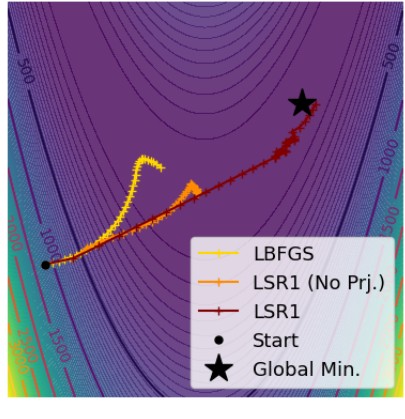
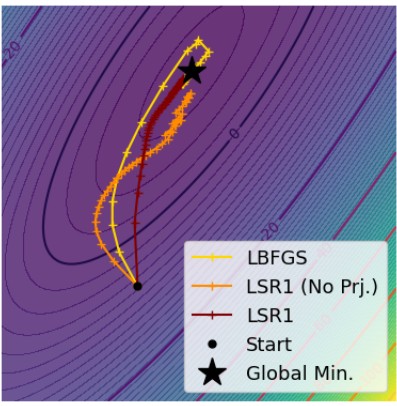

*(a)* **Rastrigin function.**     *(b)* **Rosenbrock function.**     *(c)* **Quadratic function.**

*Figure 1.* **Optimization trajectories.** Example trajectories of LBFGS, L-SR1 with PGSM, and L-SR1 without (No Prj.) on a quadratic function and two well-known challenging benchmark functions (Surjanovic & Bingham)—the Rosenbrock and Rastrigin functions. PGSM improves the performance of L-SR1 compared to the variant without PGSM (No Prj.), while maintaining model compactness.

However, with the emergence of increasingly comprehensive annotated datasets, these traditional approaches were largely supplanted by deep neural network (DNN)-based methods. These newer methods span both end-to-end regression models and iterative frameworks (Sun et al., 2023; Shin et al., 2024; Wang et al., 2025), including learned optimization (Kolotouros et al., 2019; Song et al., 2020), achieving state-of-the-art performance. Nonetheless, they come with notable trade-offs, including significantly larger model sizes and a continued reliance on vast amounts of annotated training data.

In this work, we introduce Learned-SR1 (L-SR1), a novel learned second-order optimizer that extends the classical SR1 algorithm. Our method incorporates a learnable module that generates data-driven rank-one updates to approximate the inverse Hessian, enabling more informed and efficient steps. To preserve core quasi-Newton properties, we propose a *Projection-Guided Secant Mechanism* (PGSM) that parameterizes the preconditioner through learned rank-one outer products, yielding positive semi-definite matrices by construction. During meta-training, PGSM penalizes secant mismatch in the spirit of least-squares projection onto PSD matrices, without requiring a projection subproblem at inference (Section 4.2). L-SR1 instantiates PGSM with a limited-memory buffer of these vectors and lightweight trainable modules. Building on the SR1 framework, it achieves state-of-the-art performance with a notably compact model size.

We evaluate our approach on both analytic benchmark functions (Fig. 1) and the real-world task of HMR. Through controlled analytic experiments, we study generalization across problem dimensions and the quality of the search directions produced by L-SR1, providing insight into its behavior beyond a single downstream application. On HMR, a challenging high-dimensional, non-convex optimization

problem in which each iteration involves a computationally intensive 3D-to-2D projection, L-SR1 converges efficiently and integrates robustly into the iterative optimization pipeline without requiring task-specific fine-tuning. Together, these results highlight the potential of L-SR1 as a general optimization framework that can be incorporated into a wide range of optimization-based pipelines.

We summarize our key contributions as follows:

- We propose Learned-SR1 (L-SR1), a lightweight, self-supervised learned optimizer that integrates a trainable preconditioning unit into the SR1 framework, enabling data-driven curvature estimation without the need for annotated data or supervised meta-training.

- We introduce PGSM, a projection-guided secant mechanism with a PSD-by-construction rank-one parameterization and a meta-training secant-violation penalty motivated by PSD projection, preserving core quasi-Newton structure without extra inference cost.

- We demonstrate that L-SR1 can be integrated into optimization-based HMR pipelines, where it consistently outperforms classical and learned solvers in terms of convergence and generalization.

**Conflict of Interest Disclosure.**    The authors have no financial conflicts of interest to disclose regarding this work.

## 2. Related Work

**Second-Order Optimization and Preconditioning** Second-order optimization methods leverage curvature information to accelerate convergence, typically by using the inverse Hessian matrix as a preconditioner (Boyd et al.,

2004; Bertsekas, 1999). Since computing and inverting the full Hessian is often impractical, Quasi-Newton methods such as DFP and BFGS (Bertsekas, 1999), LBFGS (Nocedal, 1980; Liu & Nocedal, 1989), and SR1 (Conn et al., 1991; Khalfan et al., 1993) were introduced to iteratively estimate this preconditioner. SR1, in particular, uses rank-one updates and demonstrated favorable convergence under mild conditions. These ideas were extended to large-scale problems in deep learning using scalable approximations (Martens & Grosse, 2015; Gupta et al., 2018; Yao et al., 2021).

**Learned Optimization and Model-Based Deep Learning**  Gradient-based methods are lightweight and broadly applicable but do not explicitly model curvature information. Classical Quasi-Newton methods, in contrast, exploit curvature structure at the cost of more expensive updates. Learned optimization seeks to bridge classical optimization principles with the adaptability of deep learning by embedding trainable components into iterative solvers.

Learned optimization methods generate adaptive update rules from data (Andrychowicz et al., 2016; Li & Malik, 2016; Metz et al., 2020; Wichrowska et al., 2017; Metz et al., 2022; 2023; Chen et al., 2022). While early works primarily focused on first-order optimization dynamics, more recent approaches introduced richer architectures, including Transformer-based optimizers (Gärtner et al., 2023) and low-rank attention mechanisms (Jain et al., 2023). In parallel, model-based deep learning integrated learning into principled algorithmic formulations to preserve interpretability and robustness (Shlezinger et al., 2020; Revach et al., 2022; Shlezinger et al., 2023).

Several works have explored learned optimizers inspired by second-order methods and Quasi-Newton optimization. (Li et al., 2020) proposed a learned optimizer motivated by Quasi-Newton principles, while (Ayad et al., 2024) developed an unrolled BFGS-like network for CT reconstruction. In addition, (Gärtner et al., 2023) introduced a general-purpose learned second-order optimizer, and (Liao et al., 2023) learned a preconditioning matrix that is updated online during optimization without explicit meta-training.

In contrast to previous work, our approach introduces PGSM—a projection-guided secant mechanism with a PSD-by-construction rank-one parameterization and a meta-training secant penalty—within a lightweight, self-supervised SR1-inspired preconditioner that stores the learned rank-one vectors in a limited-memory buffer at inference.

**Human Mesh Recovery (HMR)**  Human Mesh Recovery (HMR) aims to estimate 3D human meshes from single RGB images, a highly ill-posed problem due to the loss of depth.

Early works approached HMR through optimization-based fitting (Bogo et al., 2016; Pavlakos et al., 2019), which, while data-efficient, were slow and sensitive to initialization. With the emergence of large-scale datasets, deep learning methods were introduced, including regression-based approaches (Shin et al., 2024; Sun et al., 2023; Wang et al., 2025) and iterative refinement techniques, including learned optimization (Kolotouros et al., 2019; Song et al., 2020; Shetty et al., 2023). These models achieved impressive performance but depended heavily on annotated data and large architectures. In contrast, our method integrates a learned SR1-inspired optimizer into the HMR process, outperforming learned optimization-based methods while requiring neither large models nor explicit fine-tuning.

## 3. Theoretical Background and Preliminaries

### 3.1. Learned Optimization

A typical learned optimization framework consists of the following update rule:

$$\mathbf{x}_{k+1} \leftarrow \mathbf{x}_k + \varphi_\Theta \left( \nabla f(\mathbf{x}_k), \mathbf{x}_k, \ldots \right), \qquad (1)$$

where $\varphi_\Theta(\cdot)$ is a learnable function parameterized by $\Theta$, which can be conditioned on a variety of features, such as the current iterate and gradient.

Training a learned optimizer (*meta-training*) alternates between *inner* and *outer* iterations. In each outer iteration, the optimizer performs $K$ unrolled inner steps, whose objective values are used to compute a *meta-loss*, often of the form:

$$\mathcal{L}_{\text{meta}} = \sum_{k=1}^{K} f(\mathbf{x}_k), \qquad (2)$$

with optional additional supervision terms to leverage task-specific signals.

Gradients of this meta-loss with respect to $\Theta$ are then back-propagated through the unrolled computation graph, and the parameters are updated using a *meta-optimizer*.

### 3.2. Quasi-Newton Methods

Let $f : \mathbb{R}^n \to \mathbb{R}$ be a twice differentiable objective function. The Quasi-Newton (QN) update step is given by

$$\mathbf{x}_{k+1} \leftarrow \mathbf{x}_k - \alpha_k \mathbf{B}_k \mathbf{g}_k, \qquad (3)$$

where $\mathbf{B}_k$ is a preconditioning matrix approximating the inverse Hessian at $\mathbf{x}_k$, $\mathbf{g}_k = \nabla f(\mathbf{x}_k)$ is the gradient, and $\alpha_k$ is a step size.

QN methods differ in how they update $\mathbf{B}_k$, but they all satisfy the *secant constraint*, derived from a first-order Taylor approximation. Defining $\mathbf{p}_k = \mathbf{x}_k - \mathbf{x}_{k-1}$ and

---

**Algorithm 1** Learned-SR1 (L-SR1)

---

**Inputs:** Objective $f \in \mathcal{C}^2$, initial point $\mathbf{x}_0 \in \mathbb{R}^n$, initial gradient $\mathbf{g}_0$, buffer $\mathcal{B}_L = \{\emptyset\}$

  Initialize $\mathbf{p}_0 \leftarrow \mathbf{x}_0, \mathbf{q}_0 \leftarrow \mathbf{g}_0, \mathbf{d}_0 \leftarrow \mathbf{g}_0$

  **for** $k = 1, 2, \ldots$ **do**

    $\mathbf{f}_k \leftarrow \mathcal{E}(\mathbf{x}_{k-1}, \mathbf{p}_{k-1}, \mathbf{d}_{k-1}, \mathbf{g}_{k-1}, \mathbf{q}_{k-1})$

    $\mathbf{v}_k \leftarrow \mathcal{P}(\mathbf{f}_k), \boldsymbol{\alpha}_k \leftarrow \mathcal{G}(\mathbf{f}_k)$

    **if** $|\mathcal{B}_L| = L$ **then**

      Discard oldest element in $\mathcal{B}_L$

    **end if**

    $\mathcal{B}_L \leftarrow \mathcal{B}_L \cup \mathbf{v}_k$

    $\mathbf{d}_k \leftarrow \mathbf{g}_{k-1} + \sum_{\mathbf{v} \in \mathcal{B}_L} \mathbf{v}\mathbf{v}^\top \mathbf{g}_{k-1}$

    $\mathbf{x}_k \leftarrow \mathbf{x}_{k-1} - \boldsymbol{\alpha}_k \odot \mathbf{d}_k$

    $\mathbf{g}_k \leftarrow \nabla f(\mathbf{x}_k)$

    $\mathbf{p}_k \leftarrow \mathbf{x}_k - \mathbf{x}_{k-1}, \mathbf{q}_k \leftarrow \mathbf{g}_k - \mathbf{g}_{k-1}$

  **end for**

**Output:** $\mathbf{x}^* \leftarrow \mathbf{x}_k$

---

$\mathbf{q}_k = \mathbf{g}_k - \mathbf{g}_{k-1}$, the secant condition is

$$\mathbf{B}_k \mathbf{q}_k = \mathbf{p}_k. \tag{4}$$

This condition ensures that the preconditioner $\mathbf{B}_k$ captures local curvature information. Additionally, $\mathbf{B}_k$ is typically required to be positive semi-definite to guarantee that the update direction is a descent direction. The standard descent argument associated with Eq. (3) assumes a symmetric positive semi-definite $\mathbf{B}_k$ and a scalar step size $\alpha_k > 0$.

In L-SR1 (Section 4, Alg. 1), the matrix is symmetric positive semi-definite by construction, but the trainable update applies a coordinate-wise vector of learning rates $\boldsymbol{\alpha}_k$ to the preconditioned direction. Equivalently, the map from $\mathbf{g}_k$ to the step contains a diagonal left factor, so the composite linear operator need not be symmetric and the PSD-based intuition alone does not automatically yield a descent guarantee on the full update; practical behavior is therefore established empirically through meta-training and the experiments in Section 5.

## 4. Learned Symmetric-Rank-One (L-SR1)

L-SR1 is a learned extension of the classical Symmetric Rank-One (SR1) Quasi-Newton method, designed to integrate lightweight trainable modules into a principled second-order optimization framework. The method enhances convergence while maintaining scalability and generalization across problem dimensions.

At its core, SR1 approximates the inverse Hessian matrix using a rank-one update of the form

$$\mathbf{B}_k \leftarrow \mathbf{B}_{k-1} + \mathbf{u}_k \mathbf{v}_k^\top, \tag{5}$$

where the vectors $\mathbf{v}_k = \mathbf{p}_k - \mathbf{B}_{k-1}\mathbf{q}_k$ and $\mathbf{u}_k = \mathbf{v}_k/(\mathbf{v}_k^\top \mathbf{q}_k)$ are chosen to satisfy the secant constraint presented in Eq. (4). This ensures that the update captures local curvature information of the objective function.

A significant advantage of the SR1 structure is that it relies solely on outer products of low-dimensional vectors. This allows for a limited-memory implementation, where instead of storing the full matrix $\mathbf{B}_k$, L-SR1 maintains a fixed-size buffer $\mathcal{B}_L$ of capacity $L$ containing the most recent $L$ vectors. These vectors are used to reconstruct $\mathbf{B}_k$ implicitly during optimization. If the buffer exceeds its capacity, the oldest entries are discarded, ensuring memory efficiency in high-dimensional problems.

A central design principle in L-SR1 is invariance to the problem dimension. All learnable components are constructed to operate element-wise, ensuring that the optimizer generalizes across problem sizes without re-training.

However, a known limitation of SR1 is that its updates do not guarantee positive semi-definiteness of $\mathbf{B}_k$, which can result in non-descent directions and instability. In the learned optimization realm, a naive fix is to use outer products of the form $\mathbf{v}\mathbf{v}^\top$ as was done in (Gärtner et al., 2023), which are always positive semi-definite and symmetric, but such updates fail to satisfy the secant constraint. Traditional methods like LBFGS address this using more elaborate update strategies. In contrast, L-SR1 addresses this limitation through PGSM (Section 4.2): a PSD-by-construction rank-one parameterization together with a meta-training secant penalty motivated by projection, without solving an explicit projection at each inner step.

A summary of the proposed method is given in Alg. 1 and a block diagram is in Fig 2. We now describe the components of the L-SR1 algorithm in detail. The trainable modules are introduced in Section 4.1; PGSM is defined in Section 4.2.

### 4.1. Learned Components

To enrich SR1 with data-driven flexibility, L-SR1 integrates three neural modules following the model-based deep learning paradigm of embedding learned operations inside a classical update while preserving structure (Shlezinger et al., 2023). All modules follow a standard and lightweight multi-layer perceptron (MLP) architecture and operate element-wise over the input dimensions, ensuring compatibility with varying problem sizes (Metz et al., 2022; 2023; Chen et al., 2022). Model architectures are presented in Appendix C.1.

**Input Encoder** $\mathcal{E}$  The encoder constructs a latent representation of the optimization state. It takes as input the current point $\mathbf{x}_{k-1}$, step $\mathbf{p}_{k-1}$, descent direction $\mathbf{d}_{k-1}$, gradient $\mathbf{g}_{k-1}$, and gradient step $\mathbf{q}_{k-1}$, which are concatenated into an array of shape $N \times 5$, where $N$ is the problem dimension. The encoder then maps this input to a latent representation $\mathbf{f}_k \in \mathbb{R}^{N \times M}$, with $M > 5$, lifting the per-coordinate features into a richer representation—as in prior per-parameter learned optimizers (Gärtner et al., 2023; Metz et al., 2022; 2023)—that informs the subsequent modules.

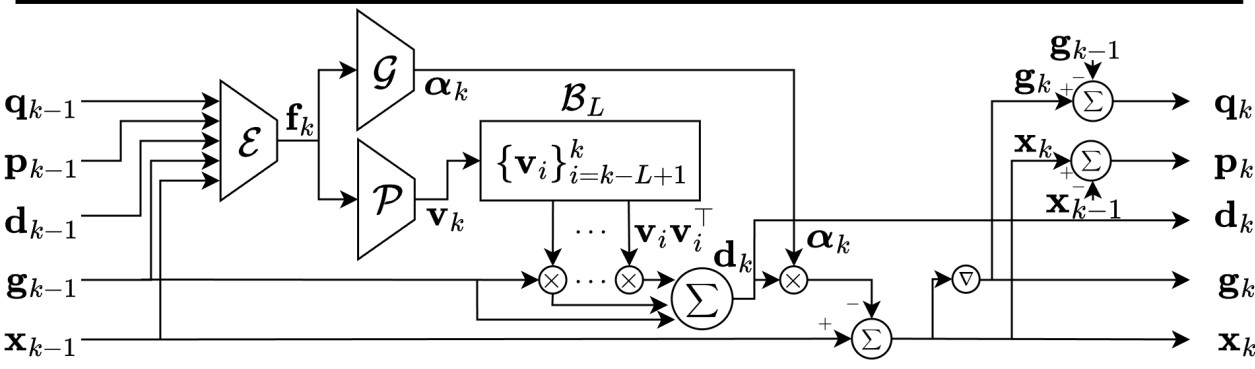

*Figure 2.* **Learned-SR1 (L-SR1) iteration block diagram.** At each iteration $k$, the Input Encoder $\mathcal{E}$ receives the vectors $\mathbf{x}_{k-1}$, $\mathbf{p}_{k-1}$, $\mathbf{d}_{k-1}$, $\mathbf{g}_{k-1}$, and $\mathbf{q}_{k-1}$, producing a feature vector $\mathbf{f}_k$. This is passed to the Vector Generator $\mathcal{P}$, which outputs a new direction vector $\mathbf{v}_k$, and to the Learning Rate Generator $\mathcal{G}$, which produces element-wise learning rates $\boldsymbol{\alpha}_k$ (Sec. 4.1)). The updated descent direction $\mathbf{d}_k$ is computed as a sum of rank-one terms $\mathbf{v}_i\mathbf{v}_i^\top \mathbf{g}_{k-1}$, using the last $L$ vectors stored in the buffer $\mathcal{B}_L$. Finally, the optimization step is performed using $\boldsymbol{\alpha}_k$ and $\mathbf{d}_k$.

**Vector Generator $\mathcal{P}$**  This module produces a single vector $\mathbf{v}_k \in \mathbb{R}^N$ given $\mathbf{f}_k$ at each iteration. Under PGSM (Section 4.2), outer products $\mathbf{v}_k\mathbf{v}_k^\top$ build a PSD preconditioner $\tilde{\mathbf{B}}_k$ and the secant penalty guides meta-training; a limited-memory buffer stores the most recent $L$ vectors for efficient inference.

**Learning Rate Generator $\mathcal{G}$**  The learning rate generator takes the latent representation from the encoder $\mathbf{f}_k$ and outputs a vector $\tilde{\boldsymbol{\alpha}}_k \in \mathbb{R}^N$, interpreted as the logarithm of coordinate-wise learning rates, following (Gärtner et al., 2023). The final learning rate vector $\boldsymbol{\alpha}_k$ is computed element-wise using the transformation:

$$\boldsymbol{\alpha}_k = \gamma_1 \cdot \exp\left(\gamma_2 \cdot \tilde{\boldsymbol{\alpha}}_k\right),$$

where $\gamma_1$ and $\gamma_2$ are fixed scalar hyperparameters (not learned). Appendix B.2 compares coordinate-wise and scalar learning-rate parameterizations, supporting this design choice.

### 4.2. Projection-Guided Secant Mechanism (PGSM)

As discussed earlier, our goal is to construct preconditioning matrices $\mathbf{B}_k$ that are both positive semi-definite (PSD) and approximately satisfy the secant equation (Eq. (4)). We motivate this objective through a projection formulation that minimizes the violation of the secant condition:

$$\mathbf{B}_k^* = \pi_+(\mathbf{B}_k) = \underset{\mathbf{B}_k \in S_+}{\operatorname{argmin}} \|\mathbf{p}_k - \mathbf{B}_k\mathbf{q}_k\|_2^2, \qquad (6)$$

where $S_+$ denotes the space of positive semi-definite matrices. This formulation seeks a PSD matrix that best approximates the secant constraint in a least-squares sense and motivates the proposed Projection-Guided Secant Mechanism (PGSM).

To ensure scalability and efficient computation, we constrain $\mathbf{B}_k$ to take the following structured form:

$$\tilde{\mathbf{B}}_k = \mathbf{B}_0 + \sum_{i=1}^{L} \mathbf{v}_i\mathbf{v}_i^\top, \qquad (7)$$

where each vector $\mathbf{v}_i \in \mathbb{R}^n$ is produced by the vector generator module $\mathcal{P}$ and stored in a fixed-size buffer of length $L$. This construction ensures that $\tilde{\mathbf{B}}_k$ remains positive semi-definite as long as $\mathbf{B}_0 \succcurlyeq \mathbf{0}$, which we initialize as the identity matrix. The use of outer products is inspired by the classical SR1 update structure and is also consistent with techniques adopted in prior learned optimization works, such as (Gärtner et al., 2023), which promote symmetry and positivity.

Within PGSM, the projection objective is formulated as the following optimization problem:

$$\tilde{\mathbf{B}}_k^* = \tilde{\pi}_+(\tilde{\mathbf{B}}_k) = \underset{\tilde{\mathbf{B}}_k = \mathbf{B}_0 + \sum_{i=1}^{L} \mathbf{v}_i\mathbf{v}_i^\top}{\operatorname{argmin}} \|\mathbf{p}_k - \tilde{\mathbf{B}}_k\mathbf{q}_k\|_2^2. \quad (8)$$

Rather than solving (8) explicitly, PGSM uses the corresponding mismatch objective as the *secant penalty* $\mathcal{R}_{\text{sec}}$, which is incorporated into the overall meta-training loss:

$$\mathcal{L}_{\text{meta}} = \frac{1}{K} \sum_{k=1}^{K} \left( f(\mathbf{x}_k) + \lambda_{\text{sec}} \cdot \|\mathbf{p}_k - \tilde{\mathbf{B}}_k\mathbf{q}_k\|_2^2 \right), \quad (9)$$

where $\lambda_{\text{sec}}$ is a hyperparameter controlling the importance of satisfying the secant condition, and $K$ is the number of unrolled optimization steps per meta-iteration.

PGSM therefore maintains PSD preconditioning matrices by construction while encouraging approximate secant consistency through meta-training; no projection subproblem

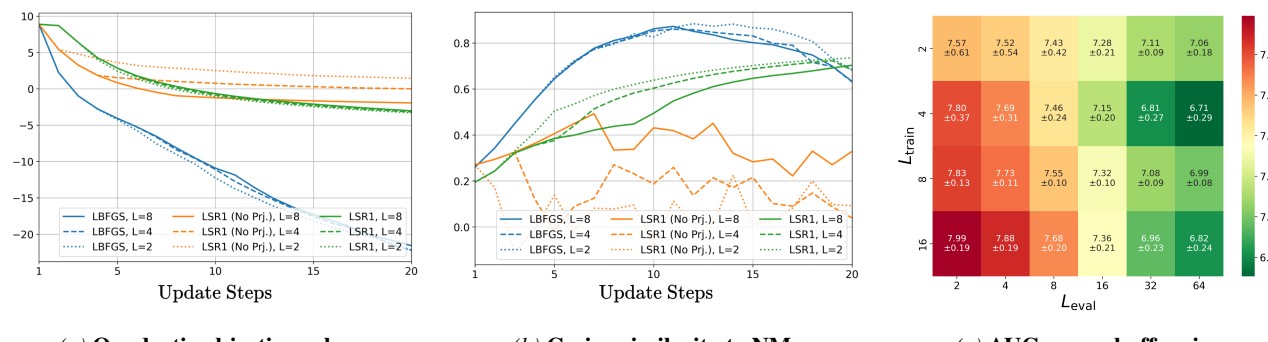

*(a)* **Quadratic objective values.**     *(b)* **Cosine similarity to NM.**     *(c)* **AUC across buffer sizes.**

*Figure 3.* **Quadratic functions experiments.** Figures 3a and 3b show results from our quadratic experiments (Section 5.1.1), comparing L-SR1 with PGSM and without (No Prj.) to LBFGS (Nocedal, 1980), which serves as a reference. Figures 3a and 3b report objective values and cosine similarities with the Newton direction, respectively, averaged over the test set. PGSM improves L-SR1 over the variant without PGSM (No Prj.) in both convergence and alignment with the Newton direction. Figure 3c shows the area under loss curve (AUC) as a function of evaluation buffer size for models trained with different training buffer sizes, averaged over different seeds. Performance improves monotonically with evaluation buffer, and remains robust across training buffers.

is solved at inference. L-SR1 applies the resulting precon­ditioner by reconstructing $\tilde{\mathbf{B}}_k$ from the limited-memory vector buffer, with no additional overhead beyond rank-one accumulation.

# 5. Experimentation

We conduct a series of experiments to evaluate both the performance and generalization properties of the proposed L-SR1 optimizer. We begin with controlled analytic experi­ments on quadratic objectives (Section 5.1.1), where we iso­late the effect of PGSM and systematically study robustness across problem dimensions and buffer sizes. These experi­ments provide insight into the optimizer's convergence be­havior and transfer to unseen problem dimensions. We then evaluate L-SR1 on a suite of benchmark optimization prob­lems (Surjanovic & Bingham) (Section 5.1.2), demonstrat­ing consistent improvements over classical and learned base­lines across diverse objective landscapes. Finally, we apply L-SR1 to a real-world 3D human mesh recovery task (Sec­tion 5.2), highlighting its effectiveness in high-dimensional, structured optimization settings; there we compare against LGD (Song et al., 2020) and an L-SR1 variant trained with­out the secant penalty, which we treat as representative learned-optimization baselines in this pipeline. Further re­sults and implementation details are given in Appendix B.

**Implementation Details.**    Our method is implemented en­tirely in PyTorch (Paszke et al., 2019)[1]. Following standard learned optimization practices (Metz et al., 2022; 2023; Chen et al., 2022), our training setup consists of inner optimization loops and outer meta-iterations. Using Py­Torch's autograd framework, we compute gradients of the

inner objective with respect to the optimization variables during each unrolled step, and gradients of the meta-loss (Eq. (9)) with respect to the optimizer's parameters dur­ing each meta-iteration. We use the AdamW (Loshchilov & Hutter, 2017) optimizer for meta-training, with a fixed learning rate of $10^{-4}$, momentum parameters $\beta_1 = 0.9$, $\beta_2 = 0.999$, and weight decay coefficient $\lambda = 0.01$. Unless otherwise stated, meta-training is run for 10K iterations and takes approximately 15 hours on a single NVIDIA GeForce RTX 3090 GPU. For non-learned baselines, learning rates were selected by grid search on each task (details in Ap­pendix C). Per-iteration runtime and memory are reported in Appendix A.

## 5.1. Analytic Experiments

### 5.1.1. QUADRATIC FUNCTIONS

We begin our evaluation with randomly generated quadratic functions, which provide a controlled and analytically well-understood setting for studying optimizer behavior. This setting allows us to isolate the effect of individual design choices in L-SR1 in a controlled manner. In particular, we use quadratic objectives to address three questions: (i) how PGSM affects optimization dynamics; (ii) whether per­formance generalizes across problem dimensions not seen during training; and (iii) how robust the optimizer is to variations in buffer size during training and inference. By answering these questions in a simple setting, we aim to build intuition for the behavior of L-SR1 before evaluating it on more complex objectives. We also compare two sepa­rately meta-trained variants, one with a scalar learning rate and one with an element-wise learning rate, on both quadrat­ics and the HMR task; results are provided in Appendix B.2 (Table 3).

---

[1]Project page and code: https://gallif.github.io/lsr1/.

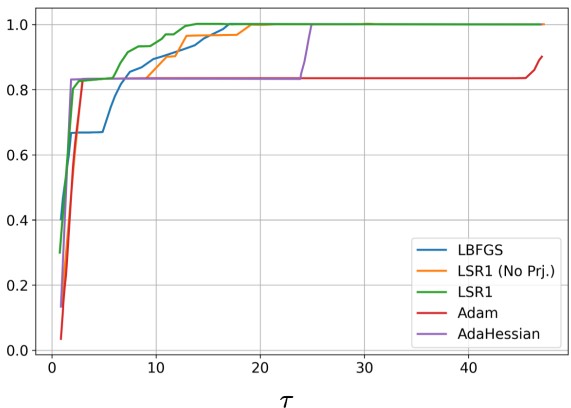

*Figure 4.* **Performance profiles** comparing L-SR1 with PGSM and without (No Prj.) to baseline optimizers on a benchmark suite of objective functions. The benchmarks consist of quadratic, Rosenbrock, and Rastrigin functions across a range of problem dimensions. L-SR1 with PGSM achieves the highest overall performance profile, indicating strong effectiveness across tasks.

We consider quadratic functions of the form

$$f(\mathbf{x}) = \frac{1}{2}\mathbf{x}^\top \mathbf{H}\mathbf{x} + \mathbf{b}^\top \mathbf{x}, \tag{10}$$

where $\mathbf{H} \in \mathbb{R}^{N \times N}$ is a random positive semi-definite matrix and $\mathbf{b} \in \mathbb{R}^N$ is a random vector. For each experiment, we generate independent collections of such objectives together with corresponding initial points $\mathbf{x}_0$.

Unless otherwise stated, meta-training and validation are performed on quadratic functions with dimension $N = 2$. A fixed validation set of 32 functions and initial points is used throughout training, while training batches consist of 128 randomly generated functions per meta-iteration. To evaluate generalization, we additionally construct a test set of 32 previously unseen functions with $N = 10$. Further implementation details are provided in the Appendix C.2.1.

**Effect of PGSM.** We first study the effect of PGSM in a controlled quadratic setting. Here all models were meta-trained using a fixed buffer size $L = 8$ and evaluated using buffer sizes $L \in \{2, 4, 8\}$. Figure 3a reports the average test loss over the first 20 optimization iterations. As expected, L-BFGS, which is data-independent and explicitly designed for quadratic problems, exhibits the fastest convergence across all tested settings. Importantly, L-SR1 with PGSM ($\lambda_{\text{sec}} > 0$) consistently outperforms the variant without PGSM (No Prj., $\lambda_{\text{sec}} = 0$), highlighting the benefit of the secant penalty during meta-training. Despite being trained in low dimension, L-SR1 with PGSM remains stable and continues to make consistent progress when evaluated on higher-dimensional objectives.

To further examine the quality of the learned updates, Fig-

ure 3b shows the average cosine similarity between the optimizer's descent directions and the corresponding Newton directions on the test set. L-SR1 with PGSM exhibits steadily increasing alignment throughout optimization, indicating that it learns curvature-aware updates that generalize beyond the training dimension. In contrast, the variant without PGSM (No Prj.) exhibits weaker and less consistent alignment, highlighting the role of the secant penalty in shaping the learned updates.

**Generalization Across Dimensions.** Appendix B.1 (Figure 10) evaluates optimizers meta-trained only on $N{=}2$ quadratics on held-out test problems at $N{=}10$, with no re-training or dimension-specific retuning. The heatmaps report mean-loss AUC over a fixed budget of 50 inner steps, swept over meta-training unroll depth and $\lambda_{\text{sec}}$. Absolute AUC is higher at $N{=}10$ than at $N{=}2$ (e.g., L-SR1 with PGSM at unroll 16 and $L{=}8$: $3.07 \pm 0.18$ vs. $7.33 \pm 0.26$ over four seeds). This is expected under a fixed iteration budget, as higher-dimensional problems require reducing more coordinates and the optimizer was never meta-trained on $N{=}10$ instances. Moreover, AUC integrates loss over the full optimization trajectory, so slower progress increases the score even when optimization remains stable, with no divergence observed in our runs. More importantly, the ranking of hyperparameter settings is largely preserved across dimensions: configurations with intermediate unroll depths that perform best at $N{=}2$ remain among the best at $N{=}10$. This suggests that the method generalizes well across dimensions, and that models trained in lower-dimensional settings transfer meaningfully to higher-dimensional problems.

**Buffer Sizes.** We evaluate the learned optimizer across training buffers $L_{\text{train}} \in \{2, 4, 8, 16\}$ and evaluation buffers $L_{\text{eval}} \in \{2, 4, 8, 16, 32, 64\}$ (Figure 3c), and report results on the test set of previously unseen problems with dimension $N = 10$. The area under the loss curve (AUC) is measured for a fixed iteration budget of 50 and averaged across random seeds. Performance consistently improves with larger evaluation buffers, showing smooth, monotonic gains and reduced variance when additional curvature information is available at inference. For any fixed evaluation buffer, differences across training buffers are small and standard deviations remain modest, with no evidence of catastrophic mismatches. Appendix A (Figs. 8 and 9) shows that meta-training incurs substantial memory cost due to unrolled graphs, which increases with buffer size and number of unrolled iterations. In contrast, inference is memory- and time-bounded with fixed per-iteration cost.

### 5.1.2. PERFORMANCE PROFILES

We compare our learned optimizer against several baselines using performance profiles, a standard evaluation method

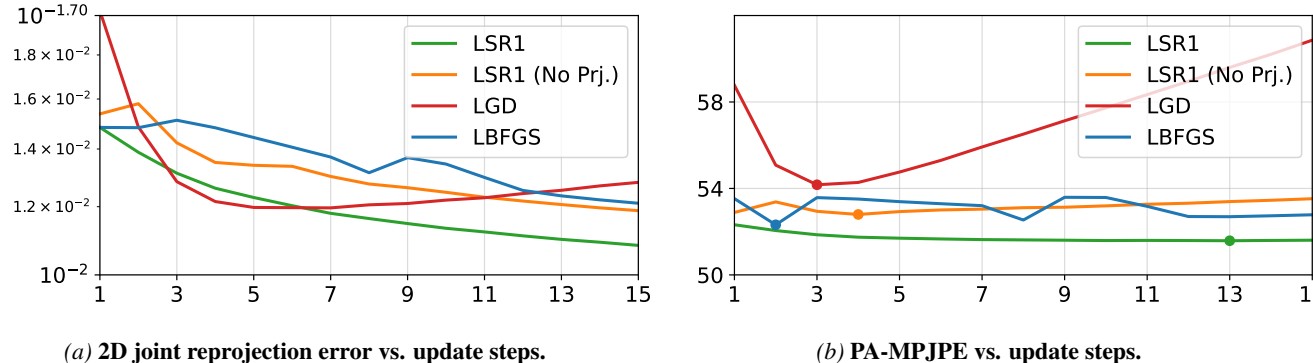

*(a)* **2D joint reprojection error vs. update steps.**   *(b)* **PA-MPJPE vs. update steps.**

*Figure 5.* **HMR error curves on 3DPW.** We compare 15 inner iterations of LGD (Song et al., 2020), L-SR1 with PGSM (ours), L-SR1 without (No Prj.), and LBFGS (Liu & Nocedal, 1989) initialized with our learned initialization. Fig. 5a shows 2D joint reprojection error; Fig. 5b shows PA-MPJPE. In Fig. 5b, markers indicate the best error achieved within these 15 inner iterations for each method. While LGD briefly achieves lower 2D error, L-SR1 with PGSM attains the strongest 3D accuracy, consistently outperforms the No Prj. variant, and continues to improve across iterations.

for optimization algorithms (Dolan & Moré, 2002; Sergeyev & Kvasov, 2015; Beiranvand et al., 2017; Chen et al., 2022; Gärtner et al., 2023). Protocol details (definitions, objective suite, and hyperparameters) are given in Appendix C.2.2. We meta-train on random quadratics and on Rosenbrock and Rastrigin functions with $N{=}100$, and evaluate on a 30-problem suite spanning quadratics and non-convex objectives with dimensions 50–1000. We compare L-SR1 (with and without PGSM) to L-BFGS (Nocedal, 1980), Adam (Kingma & Ba, 2014), and AdaHessian (Yao et al., 2021), reporting performance after a fixed iteration budget $K$; per-iteration costs differ across solvers (Appendix A). Figure 4 shows that L-SR1 with PGSM achieves the highest performance profile.

### 5.2. Monocular Human Mesh Recovery (HMR)

We integrate L-SR1 into the learned optimization–based HMR framework of (Song et al., 2020), which combines a trainable initialization with gradient-descent–inspired update modules. Self-supervision during training replaces the need for manually engineered loss terms (Bogo et al., 2016; Pavlakos et al., 2019). Using L-SR1 demonstrates efficient convergence and improved accuracy compared to existing optimization-based HMR pipelines, while requiring no task-specific fine-tuning and using a smaller model. HMR provides a high-dimensional, non-convex real-world scenario to study optimizer behavior in practice (see Appendix B.4 for reduced-data experiments). In our evaluation, LGD and L-SR1 without PGSM represent the main learned-optimization baselines: LGD follows the original learned first-order-style inner loop of this framework, while the latter uses the same L-SR1 parameterization but omits the secant penalty at meta-training ($\lambda_{\text{sec}} = 0$; labeled *w/o sec.* or No Prj. in tables and figures).

**Training**   Our model follows the training scheme of (Song et al., 2020) and is trained exclusively on the AMASS dataset (Mahmood et al., 2019), which contains approximately 20M human meshes with ground-truth SMPL shape and pose parameters ($\boldsymbol{\beta}_{\text{gt}}, \boldsymbol{\theta}_{\text{gt}}$). Using the SMPL body model (Loper et al., 2023)[2], we synthetically generate 3D meshes and joint locations, which are projected to obtain 2D keypoints $\mathbf{x}_{\text{gt}}$. These synthetic 2D and 3D annotations provide the self-supervised training signals. At each inner iteration $k$, the optimizer estimates parameters ($\boldsymbol{\beta}_k, \boldsymbol{\theta}_k$), from which predicted 3D and 2D joints ($\mathbf{X}_k, \mathbf{x}_k$) are obtained. The inner objective is defined as the weighted 2D reprojection loss

$$f_{\text{rec}}\left(\mathbf{x}_k\right) = \left\|\mathbf{w} \odot \left(\mathbf{x}_k - \mathbf{x}_{\text{gt}}\right)\right\|_1, \tag{11}$$

where $\mathbf{w}$ are confidence weights randomly sampled from a Bernoulli distribution during training, following (Song et al., 2020).

Incorporating the self-supervision strategy of (Song et al., 2020) together with the secant penalty, the total meta-loss is

$$\mathcal{L}_{\text{meta}} = \sum_{k=1}^{K} \left(\lambda_{\text{2D}} f\left(\mathbf{x}_k\right) + \lambda_{\text{self}}\|\Theta_k - \Theta_{\text{gt}}\|_1\right) + \lambda_{\text{sec}}\mathcal{R}_{\text{sec}},$$

(12)

where $\Theta_k = \{\mathbf{X}_k, \boldsymbol{\theta}_k, \boldsymbol{\beta}_k\}$ and $\lambda_{\text{2D}}, \lambda_{\text{self}}$, and $\lambda_{\text{sec}}$ are scalar hyperparameters. Further implementation details are provided in Appendix C.2.3.

#### 5.2.1. EVALUATION ON 3DPW

We evaluate our method on the challenging 3DPW dataset (von Marcard et al., 2018), which features complex, in-the-wild poses. We report PA-MPJPE on the test set, following

---

[2]Although more expressive body models exist (Pavlakos et al., 2019; Osman et al., 2020), SMPL is adopted for its widespread use and ease of integration.

*Table 1.* **Evaluation on 3DPW.** PA-MPJPE of regression-based and optimization-based methods, with total inference time reported for learned optimization methods (timing details in Appendix A). Most approaches are trained on large multi-dataset combinations including in-the-wild data, while some methods are trained exclusively on AMASS (Mahmood et al., 2019). Methods marked with † use 3DPW as part of their training data. L-SR1$_{4\text{-steps}}$ reports PA-MPJPE at the 4th inner iteration for comparison with LGD. L-SR1$_{w/o\ sec.}$ is a variant trained without the secant term.

| | Method | PA-MPJPE | AMASS only | Time (ms) |
|---|---|---|---|---|
| Regres. | TRACE† (Sun et al., 2023) | 50.8 | ✗ | – |
| | WHAM (ViT)† (Shin et al., 2024) | 35.9 | ✗ | – |
| | PromptHMR† (Wang et al., 2025) | 36.6 | ✗ | – |
| Optimization | SMPLify (Bogo et al., 2016) | 106.80 | No training | – |
| | SPIN (Kolotouros et al., 2019) | 59.20 | ✗ | – |
| | LGD (Song et al., 2020) | 55.90 | ✓ | 664 |
| | L-SR1$_{4\text{-steps}}$ (Ours) | 51.74 | ✓ | **364** |
| | L-SR1$_{w/o\ sec.}$ (Ours) | 52.80 | ✓ | 364 |
| | **L-SR1$_{opt\text{-}horizon}$ (Ours)** | **51.58** | ✓ | 1183 |

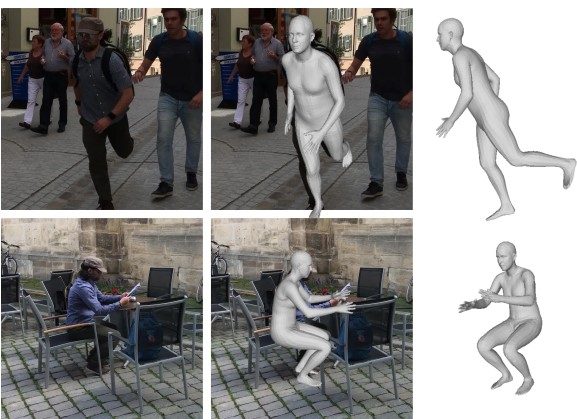

*Figure 6.* **Qualitative examples from 3DPW.**

the evaluation protocol of (Song et al., 2020), and using the 2D keypoints provided by OpenPose (Cao et al., 2019). Table 1 summarizes PA-MPJPE, training-data scope, and total inference time on 3DPW. L-SR1 outperforms LGD in accuracy with a smaller model and lower wall-clock cost at the 4-step horizon; the opt-horizon configuration trades additional inference time for the best reported PA-MPJPE. Fig. 5 shows the corresponding 2D reprojection inner loss and PA-MPJPE curves over optimization iterations. A full per-iteration runtime and memory breakdown is given in Appendix A. Qualitative examples are shown in Fig. 6, with additional examples available in Appendix E.

### 5.2.2. SECANT PENALTY AT INFERENCE

Recall the secant penalty (Eq. (9)) given in Section 4.2. The corresponding difference terms are denoted as $\mathbf{p}_k = \mathbf{x}_k - \mathbf{x}_{k-1}$ and $\mathbf{q}_k = \mathbf{g}_k - \mathbf{g}_{k-1}$, where $\mathbf{x}_k$ and $\mathbf{g}_k$ are the inner-loop parameters and gradients, respectively.

In HMR, $\mathbf{x}_k$ includes SMPL parameters, in particular the

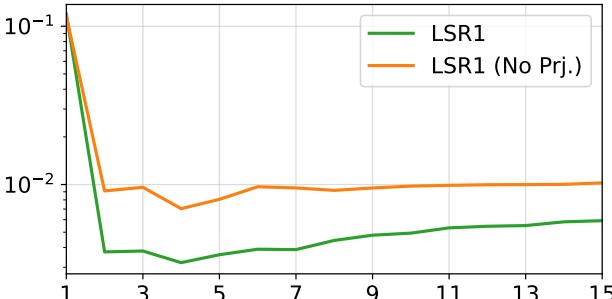

*Figure 7.* **Pose-block secant penalty on HMR inner iterations (3DPW).** Average $r_k^{(\boldsymbol{\theta})}$ from (13) over the 3DPW test set versus inner-update step $k$ for L-SR1 with PGSM ($\lambda_{\text{sec}} > 0$) and without (No Prj.). The vertical axis uses a log scale; only the first 15 inner iterations are shown, matching Fig. 5.

pose $\boldsymbol{\theta}_k$ introduced above. Let $[\cdot]_{\boldsymbol{\theta}}$ denote the subvector corresponding to the pose block. We define the *pose-block secant penalty* at iteration $k$ as

$$r_k^{(\boldsymbol{\theta})} = \left\| \left[ \mathbf{p}_k - \tilde{\mathbf{B}}_k \mathbf{q}_k \right]_{\boldsymbol{\theta}} \right\|_2^2. \qquad (13)$$

Fig. 7 reports $r_k^{(\boldsymbol{\theta})}$ averaged over the 3DPW test set for the first 15 inner iterations. We focus on the $\boldsymbol{\theta}$ block as it is the highest-dimensional and most nonlinear component of the optimization, whereas the shape parameters $\boldsymbol{\beta}_k$ are comparatively low-dimensional. We evaluate the same checkpoints as in Fig. 5: L-SR1 with PGSM ($\lambda_{\text{sec}} > 0$) and without (No Prj.). PGSM training yields substantially lower $r_k^{(\boldsymbol{\theta})}$, indicating improved secant consistency at test time.

## 6. Conclusions

We introduced L-SR1, a learned second-order optimizer built around PGSM, combining PSD-by-construction rank-one updates with secant-guided meta-training while incurring no additional projection cost at inference. On analytic benchmarks, L-SR1 with PGSM achieves the strongest overall performance profile, while on HMR it improves convergence and accuracy over competing learned optimization approaches using a smaller model and no task-specific fine-tuning. Our experiments focus on optimization-driven settings where each inner step is computationally expensive, such as HMR, where curvature-aware learned updates can reduce the number of costly iterations rather than serving as a drop-in optimizer for large-scale deep network training. More broadly, L-SR1 can accelerate iterative gradient-based frameworks by reducing the number of optimization steps required for convergence.

Limitations, including higher per-step cost than first-order methods and the scope of the empirical evaluation, are discussed in Appendix D.

## Acknowledgements

This study was funded partially by a scholarship from the Center for AI and Data Science at Tel Aviv University (TAD).

## Impact Statement

This paper presents work whose goal is to advance the field of Machine Learning. There are many potential societal consequences of our work, none which we feel must be specifically highlighted here.

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

# A. Computational Analysis

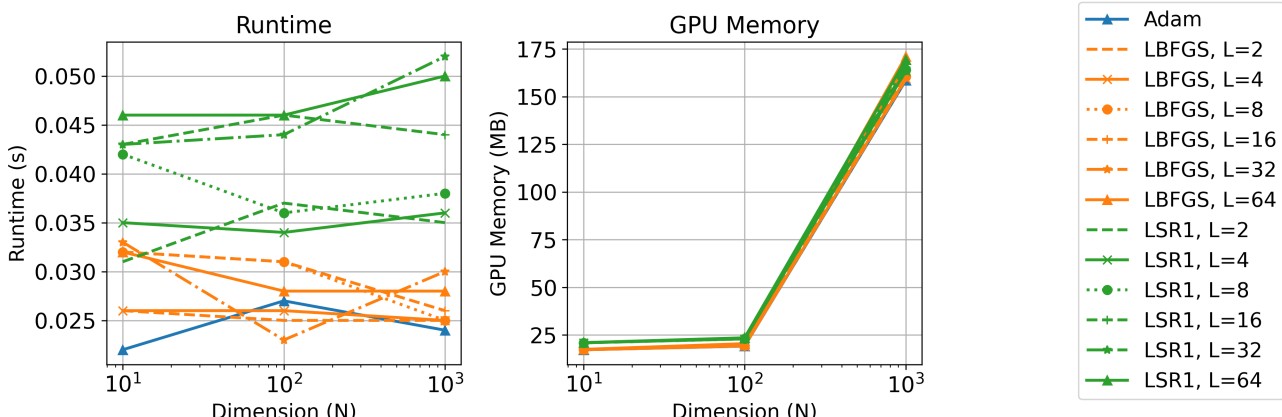

*Figure 8.* **Computational effort during inference.** Measurements were collected on a single NVIDIA RTX 3090 GPU and correspond to the mean runtime per inner inference iteration and peak memory with a batch size of 32.

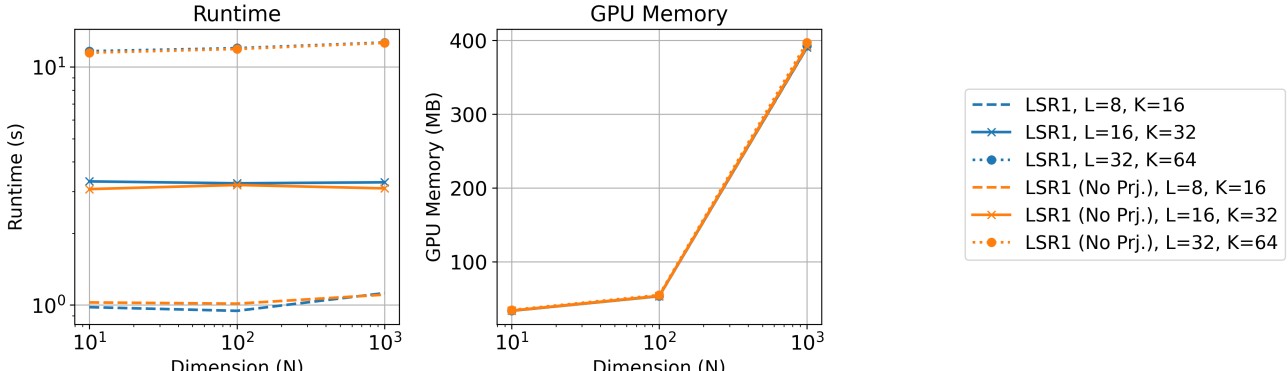

*Figure 9.* **Computational effort during meta-training.** Measurements were collected on a single NVIDIA RTX 3090 GPU and correspond to the mean runtime per inner inference iteration and peak memory usage with a batch size of 4.

*Table 2.* **HMR runtime and memory comparison.** Measurements were collected on a single NVIDIA RTX 3090 GPU and correspond to the mean runtime per inner inference iteration and peak memory usage, using a batch size of 256 and buffer size $L = 4$ for L-SR1.

| Method | Runtime (ms) | Memory (GiB) | # Params |
|---|---|---|---|
| LGD (Song et al., 2020) | 166 | 17.81 | 17.4 M |
| L-SR1 | 91 | 14.60 | 10.4 M |

# B. Further Studies and Analyses

## B.1. Generalization Across Dimensions

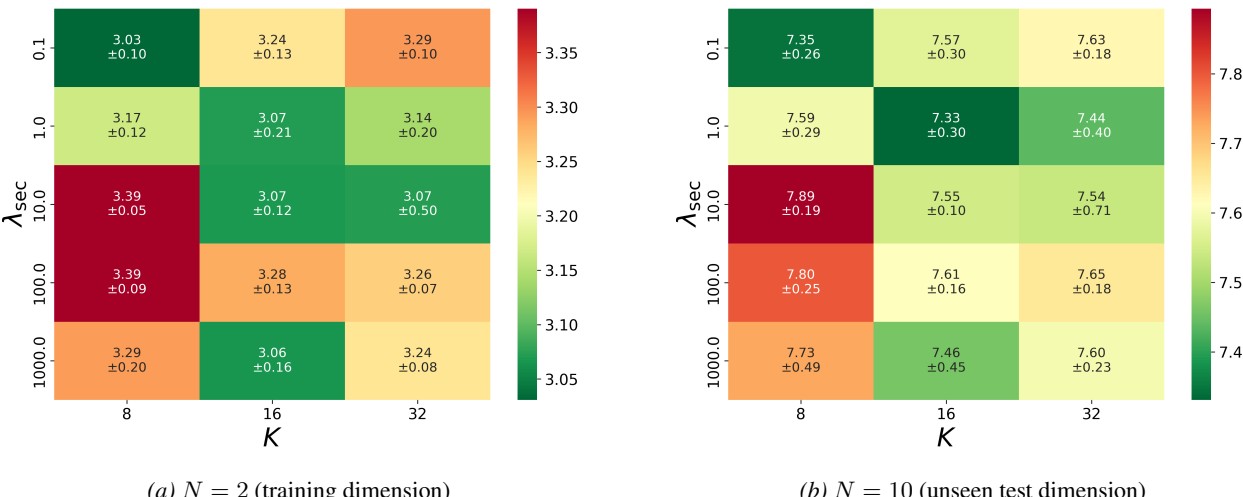

*(a) $N = 2$ (training dimension)*     *(b) $N = 10$ (unseen test dimension)*

*Figure 10.* **Generalization across dimensions.** Heatmaps report the area under the loss curve (AUC) for the learned optimizer across combinations of unrolling depth and PGSM secant-penalty weight $\lambda_{\text{sec}}$. The optimizer is meta-trained exclusively on quadratic objectives with $N = 2$ and evaluated without re-training on unseen test problems with $N = 2$ (left) and $N = 10$ (right). Mean-loss AUC is higher at $N = 10$ under a fixed 50-step budget, but hyperparameter rankings (e.g., intermediate unroll depths) are largely preserved across dimensions.

This appendix studies whether L-SR1 meta-trained on low-dimensional quadratics can be applied at a higher test dimension without re-training.

**Experimental setup.**     All models are meta-trained on randomly generated quadratic objectives with $N = 2$ (Section 5.1.1) and evaluated on held-out quadratics with $N = 10$. No re-training or dimension-specific hyperparameter tuning is performed at test time. We sweep meta-training unroll depth and the PGSM secant-penalty weight $\lambda_{\text{sec}}$, with four random seeds per configuration. Performance is summarized by the area under the mean loss curve (AUC) over 50 inner optimization steps.

**Results.**     Figure 10 compares the resulting heatmaps at $N = 2$ (training dimension) and $N = 10$ (unseen test dimension). Absolute AUC increases at $N = 10$ under the fixed step budget: with more variables per problem and meta-training confined to $N = 2$, the optimizer typically reaches higher time-averaged loss within 50 steps, even though trajectories remain well-behaved and we observe no systematic divergence. The qualitative picture across hyperparameters is nevertheless similar: intermediate unroll depths outperform very short or very long unrolls at both dimensions, so configurations that work well after meta-training at $N = 2$ remain competitive at $N = 10$ without dimension-specific retuning.

## B.2. Learning-Rate Scalarization

L-SR1 is defined with an element-wise rate vector $\boldsymbol{\alpha}_k$ from $\mathcal{G}$ (Section 4.1), and all primary experiments use that design. Here we isolate how much coordinate-wise flexibility matters by meta-training **distinct** checkpoints under an alternative inner loop: the same $\mathcal{G}$ is used, but after

$$\boldsymbol{\alpha}_k = \gamma_1 \cdot \exp\left(\gamma_2 \cdot \tilde{\boldsymbol{\alpha}}_k\right),$$

we replace $\boldsymbol{\alpha}_k$ with its coordinate average, broadcast to all parameters,

$$\boldsymbol{\alpha}_k \;\leftarrow\; \left(\frac{1}{N} \sum_{j=1}^{N} (\boldsymbol{\alpha}_k)_j\right) \mathbf{1}_N, \tag{14}$$

*inside* the unrolled inner problem, so outer meta-gradients see the reduction throughout training. This is *not* "freeze an element-wise checkpoint and insert (14) only at evaluation": the scalarized row in Table 3 comes from its own AMASS (or quadratic) meta-training run with the flag enabled end-to-end.

*Table 3.* **Element-wise versus scalarized meta-training of $\alpha_k$.**

| Training recipe | Quadratic | | 3DPW |
|---|---|---|---|
| | $N{=}2$ | $N{=}10$ | |
| Scalar | $-2.27 \pm 0.19$ | $-0.86 \pm 0.31$ | 51.80 |
| Vector | $3.07 \pm 0.21$ | $7.33 \pm 0.30$ | 51.58 |

Table 3 compares these recipes under matched settings. Each row is a separately meta-trained checkpoint; scalarized runs apply (14) throughout meta-training and evaluation. On quadratics we use unrolling depth 16, secant weight $\lambda_{\text{sec}} = 1$, fixed rate scaling $(\gamma_1, \gamma_2) = (0.4, 10^{-3})$, and buffer size $L = 8$, with four seeds on the $N{=}2$ validation set and $N{=}10$ test set; entries report mean inner loss over the first 50 unrolled steps (mean$\pm$std). On 3DPW we evaluate two AMASS-trained checkpoints with buffer $L = 4$ and $\lambda_{\text{sec}} = 1$, following the same protocol as Table 1: best PA-MPJPE (mm) over 50 inner steps. HMR serves here as a difficult, highly non-convex downstream example in contrast to the controlled quadratics: on the quadratic surrogate, scalarized meta-training attains lower mean inner loss, whereas on 3DPW element-wise meta-training achieves better PA-MPJPE by about $0.22\,\text{mm}$. We read this as evidence that difficult, structured objectives benefit from coordinate-wise learning-rate adaptation, while the simple quadratic meta-task favors a uniform step. This coordinate-wise parameterization is consistent with prior learned optimizers (Gärtner et al., 2023; Metz et al., 2022).

### B.3. Inputs and Hidden Dimension Ablations

*Table 4.* **Varying hidden dimension and encoder inputs.** Quadratic validation set errors for different configurations of the hidden dimension and selected inputs to the encoder.

| Hidden dim. | Encoder Inputs | | | | | Valid. loss |
|---|---|---|---|---|---|---|
| $d_{\text{hidden}}$ | $\mathbf{x}_{k-1}$ | $\mathbf{p}_{k-1}$ | $\mathbf{d}_{k-1}$ | $\mathbf{g}_{k-1}$ | $\mathbf{q}_{k-1}$ | |
| 128 | ✓ | ✗ | ✗ | ✓ | ✗ | -2.09 |
| 128 | ✓ | ✗ | ✓ | ✓ | ✗ | -2.16 |
| 128 | ✓ | ✓ | ✗ | ✓ | ✓ | -2.51 |
| 64 | ✓ | ✓ | ✓ | ✓ | ✓ | -1.90 |
| **128** | ✓ | ✓ | ✓ | ✓ | ✓ | **-2.56** |

We conducted a series of ablation studies on our model components. All experiments were carried out on the validation set, which comprises 32 quadratic functions as detailed in Appendix C.2.1. Table 4 summarizes the results. We evaluated the impact of varying the hidden dimension as well as different combinations of inputs to the encoder. The selected configuration is highlighted in bold.

### B.4. Training with Limited Data

To assess generalization beyond the training data, we trained L-SR1 on fractions of the AMASS dataset (80%, 10%, 1% and 0.1%) and evaluated on the full 3DPW test set. Even when trained on only 10% of the data, L-SR1 achieved a test error of 53.03, which still outperforms LGD trained on the full dataset (55.90). With only 1% of the data, the error is 56.14. Results are summarized in Table 5.

*Table 5.* **Data efficiency study.** L-SR1 trained on fractions of AMASS and evaluated on 3DPW.

| Fraction of Data | 80% | 10% | 1% | 0.1% |
|---|---|---|---|---|
| **PA-MPJPE** | 51.67 | 53.03 | 56.14 | 70.78 |

## C. Implementation Details

### C.1. L-SR1 Modules Architectures

Our proposed L-SR1 model comprises three learnable modules: an Input Encoder $\mathcal{E}$, a Vector Generator $\mathcal{P}$, and a Learning Rate (LR) Generator $\mathcal{G}$.

All modules operate element-wise and share a common MLP-based architecture, detailed in Tables 6 and 7. Concretely, inputs to each module are tensors of shape $B \times N \times d_{\text{in}}$, where $B$ denotes the batch size and $N$ the problem dimensionality. The MLP is applied independently to each coordinate: all linear layers act exclusively on the feature dimension $d_{\text{in}}$, with parameters shared across the $N$ dimensions. As a result, no mixing across coordinates occurs within the MLP itself.

Batch normalization is applied along the feature dimension by temporarily permuting the input to shape $B \times d_{\text{in}} \times N$, applying `BatchNorm1d`($d_{\text{in}}$), and permuting the tensor back. This ensures consistent normalization across batch elements and coordinates while preserving the element-wise structure of the computation.

We set $d_{\text{hidden}} = 128$ in all experiments. As described in Section 4.1 and justified in Appendix B.3, the Input Encoder uses $d_{\text{in}} = 5$ and $d_{\text{out}} = d_{\text{hidden}}$, while both the Vector and LR Generators use $d_{\text{in}} = d_{\text{hidden}}$ and $d_{\text{out}} = 1$.

*Table 6.* **MLP architecture**

| Layer | Type | Parameters |
|---|---|---|
| fc1 | Linear | Input: $d_{\text{in}}$, Output: $d_{\text{hidden}}$ |
| bn1 | BatchNorm | Features: $d_{\text{hidden}}$ |
| prelu | PReLU | – |
| do1 | Dropout | – |
| MLP1 | Basic Block | Features: $d_{\text{hidden}}$ |
| MLP2 | Basic Block | Features: $d_{\text{hidden}}$ |
| fc2 | Linear | Input: $d_{\text{hidden}}$, Output: $d_{\text{out}}$ |

*Table 7.* **Basic Block Architecture**

| Layer | Type | Parameters |
|---|---|---|
| fc1 | Linear | Input: $d_{\text{in}}$, Output: $d_{\text{hidden}}$ |
| bn1 | BatchNorm | Features: $d_{\text{hidden}}$ |
| prelu | PReLU | – |
| do1 | Dropout | – |
| fc2 | Linear | Input: $d_{\text{hidden}}$, Output: $d_{\text{hidden}}$ |
| bn2 | BatchNorm | Features: $d_{\text{hidden}}$ |
| do2 | Dropout | – |

### C.2. Experimental setup

#### C.2.1. QUADRATIC FUNCTIONS

**Data** We generate random quadratic functions of the form

$$f(\mathbf{x}) = \frac{1}{2}\mathbf{x}^\top \mathbf{H}\mathbf{x} + \mathbf{b}^\top \mathbf{x}, \tag{15}$$

where $\mathbf{H} \in \mathbb{R}^{N \times N}$ is a positive semi-definite matrix and $\mathbf{b} \in \mathbb{R}^N$ is a random vector.

To construct $\mathbf{H}$, we draw a matrix $\mathbf{A} \in \mathbb{R}^{N \times N}$ from a standard normal distribution and define

$$\mathbf{H} = \mathbf{A}^\top \mathbf{A}, \tag{16}$$

ensuring positive semi-definiteness. We compute the condition number of each $\mathbf{H}$ and discard those exceeding 1000. This process is repeated until full validation and test batches are acquired. Each matrix is then normalized to have unit Frobenius norm. Independently, $\mathbf{b}$ is sampled from a standard normal distribution and normalized to have unit Euclidean norm.

During training batch generation, we relax the condition number constraint and accept all generated matrices, regardless of their conditioning. We use $N = 2$ for both training and validation, and $N = 10$ for testing. Training batches consist of 128 samples, while validation and test sets each contain 32 samples.

**Hyperparameters** We use the AdamW optimizer (Loshchilov & Hutter, 2017) for meta-training, with a fixed learning rate of $10^{-4}$, momentum parameters $\beta_1 = 0.9$ and $\beta_2 = 0.999$, and a weight decay coefficient of $\lambda = 0.01$. Training is conducted for 10,000 meta-iterations.

We set the buffer size to $L = 8$ and the number of unrolled iterations to $K = 16$. The Learning Rate Generator is configured with scaling parameters $\gamma_1 = 0.4$ and $\gamma = 0.001$. A secant constraint is applied with a weighting factor of $\lambda_{\text{sec}} = 100$. The learning rates for non-trainable optimizers were selected through hyperparameter tuning.

### C.2.2. PERFORMANCE PROFILES

**Protocol** We use performance profiles (Dolan & Moré, 2002) to aggregate solver performance across a set of benchmark problems. Let $P$ denote the set of problems and $S$ the set of solvers. For each solver $s \in S$ and problem $p \in P$, we define the performance measure

$$m_{p,s} = \frac{\|\hat{\mathbf{x}}_{p,s} - \mathbf{x}^*\|_2}{\|\mathbf{x}_w - \mathbf{x}^*\|_2},$$

where $\hat{x}_{p,s}$ is the solution found by solver $s$ on problem $p$ after $K$ steps, $\mathbf{x}_w$ is the worst solution among solvers, and $\mathbf{x}^*$ is the global optimum. The performance ratio is

$$r_{p,s} = \frac{m_{p,s}}{\min\left(m_{p,s} \; : \; s \in S\right)},$$

with the best solver achieving $r_{p,s} = 1$. The performance profile of solver $s$ is then

$$\rho_s(\tau) = \frac{1}{|P|} \text{size}\left(\{p \in P \; : \; r_{p,s} \leq \tau\}\right), \tag{17}$$

which measures the fraction of problems on which solver $s$ is within a factor $\tau$ of the best.

**Data** The quadratic functions used in this experiment are a subset of those defined in Eq. 15, where $\mathbf{b} = \mathbf{0}$ and $\mathbf{H}$ is constrained to be diagonal. Our objective set comprises four such quadratic functions with condition numbers 1, 100, 1000, and 10000, along with the Rosenbrock and Rastrigin functions (Surjanovic & Bingham). Each function is evaluated at input dimensions $N = 50, 100, 250, 500$, and 1000, yielding a total of 30 distinct optimization problems.

The solver set includes our proposed L-SR1 optimizer, both with and without PGSM, along with three non-trainable baselines: L-BFGS (Nocedal, 1980), Adam (Kingma & Ba, 2014), and AdaHessian (Yao et al., 2021), totaling six solvers. The learning rates for non-trainable optimizers were selected through hyperparameter tuning. L-BFGS was used with default settings, without an explicit line search.

Each trainable optimizer is meta-trained on three distinct tasks: a randomly generated quadratic function (as defined in Eq. 15), the Rosenbrock function, and the Rastrigin function. For each task, a validation set of 8 fixed initial points is created and remains unchanged throughout meta-training. During each meta-iteration, a new training batch of 8 initial points is generated. Both training and validation are performed with $N = 100$.

**Hyperparameters** We use the AdamW optimizer (Loshchilov & Hutter, 2017) for meta-training, with a fixed learning rate of $10^{-4}$, momentum parameters $\beta_1 = 0.9$ and $\beta_2 = 0.999$, and a weight decay coefficient of $\lambda = 0.01$. Training is conducted for 10,000 meta-iterations. Used hyperparameters are summarized in Table 8.

*Table 8.* **Hyperparameters used in Performance profiles**

| Parameter | Quadratics | | Rosenbrock | | Rastrigin | |
|---|---|---|---|---|---|---|
| | Train | Test | Train | Test | Train | Test |
| LR gen. param. $\gamma_1$ | 0.1 | 0.1 | 0.1 | 0.1 | 0.1 | 0.1 |
| LR gen. param. $\gamma_2$ | 0.001 | 0.001 | 0.001 | 0.001 | 0.001 | 0.001 |
| Buffer size $L$ | 16 | 64 | 32 | 64 | 32 | 64 |
| Unrolled iterations $K$ | 32 | - | 64 | - | 64 | - |
| Secant loss weight $\lambda_{\text{sec}}$ | 10 | - | 1 | - | 1 | - |

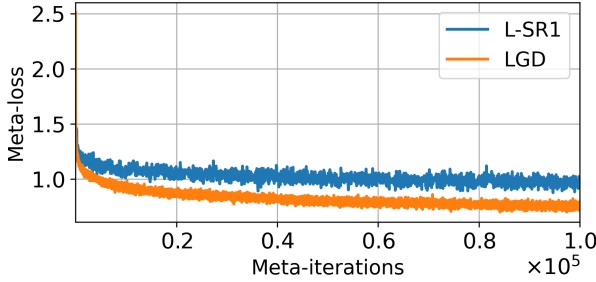
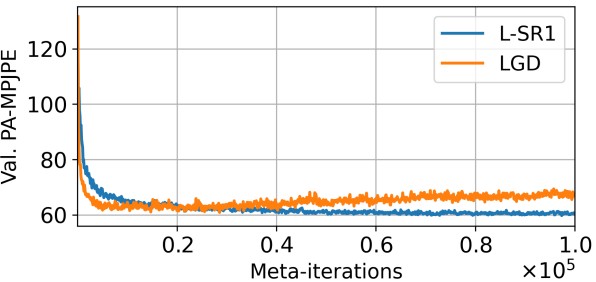

*(a)* **HMR meta-loss on AMASS (Mahmood et al., 2019).**   *(b)* **HMR validation error on 3DPW (von Marcard et al., 2018).**

*Figure 11.* **HMR meta-training on AMASS and validation erros on 3DPW.** Shown first 100K iterations. Our meta-loss is higher as it has the secant loss added to it.

### C.2.3. MONOCULAR HUMAN MESH RECOVERY (HMR)

**Data**   We follow the training and evaluation protocol of (Song et al., 2020). Training is performed on the AMASS dataset (Mahmood et al., 2019)[3], which comprises SMPL body models (Loper et al., 2023) parameterized by $(\beta_{\text{gt}}, \theta_{\text{gt}})$. At each training iteration, a batch of 2D joints is generated by projecting the corresponding 3D joints onto a randomly sampled camera view.

Validation and testing are conducted on the 3DPW dataset (von Marcard et al., 2018) using the same protocol as (Song et al., 2020), utilizing the provided 2D joint detections obtained via OpenPose (Cao et al., 2019). During training, we evaluate our model on the official 3DPW validation set and retain the checkpoint that achieves the best validation performance. The results reported in the paper are obtained by evaluating this best-performing model on the official 3DPW test set. Meta-training and validation graphs are given in Fig. 11.

**Hyperparameters**   We meta-train for 400K iterations using the AdamW optimizer (Loshchilov & Hutter, 2017), with an initial learning rate of $10^{-3}$, which is decayed by a factor of $\gamma = 0.8$ every 20K iterations. The momentum parameters are set to $\beta_1 = 0.9$ and $\beta_2 = 0.999$, and the weight decay coefficient is $\lambda = 0.01$.

We configure the meta-optimization process with a buffer size of $L = 4$ and $K = 8$ unrolled iterations. The Learning Rate Generator is parameterized with scaling factors $\gamma_1 = 0.1$ and $\gamma = 0.001$. The secant constraint is weighted with $\lambda_{\text{sec}} = 1$.

## D. Limitations

Our evaluation is intentionally focused: controlled quadratics and performance profiles on standard analytic objectives, plus HMR as a challenging non-convex downstream task. We do not claim universal superiority over every optimizer on every problem class within this scope.

---

[3]The AMASS (Mahmood et al., 2019) dataset an aggregation of the following datasets (Advanced Computing Center for the Arts and Design; Helm et al., 2015; Ghorbani et al., 2020; Troje, 2002; Carnegie Mellon University; Aristidou et al., 2019; Bogo et al., 2017; Ltd.; Taheri et al., 2020; Brahmbhatt et al., 2019; Müller et al., 2007; Chatzitofis et al., 2020; Sigal et al., 2010; Mandery et al., 2015; 2016; Krebs et al., 2021; Loper et al., 2014; Tripathi et al., 2023; Akhter & Black, 2015; University & of Singapore; Ghorbani & Black, 2021; Hoyet et al., 2012; Trumble et al., 2017; Li et al., 2024).

L-SR1 carries higher per-iteration cost than first-order methods such as Adam (Kingma & Ba, 2014) (Appendix A). Reported HMR timings count full inner iterations, including SMPL forward kinematics and rendering; fewer iterations can still yield favorable wall-clock when the inner objective dominates, but the trade-off is task-dependent.

The learned modules are meta-trained on specific task distributions (random quadratics and, for HMR, AMASS). Transfer to markedly different objectives may require re-training. Finally, coordinate-wise learning rates and limited-memory curvature improve difficult problems in our study, but they add hyperparameters (buffer size, unrolling depth, and $\lambda_{\text{sec}}$) that must be set for new settings.

# E. HMR Qualitative Results

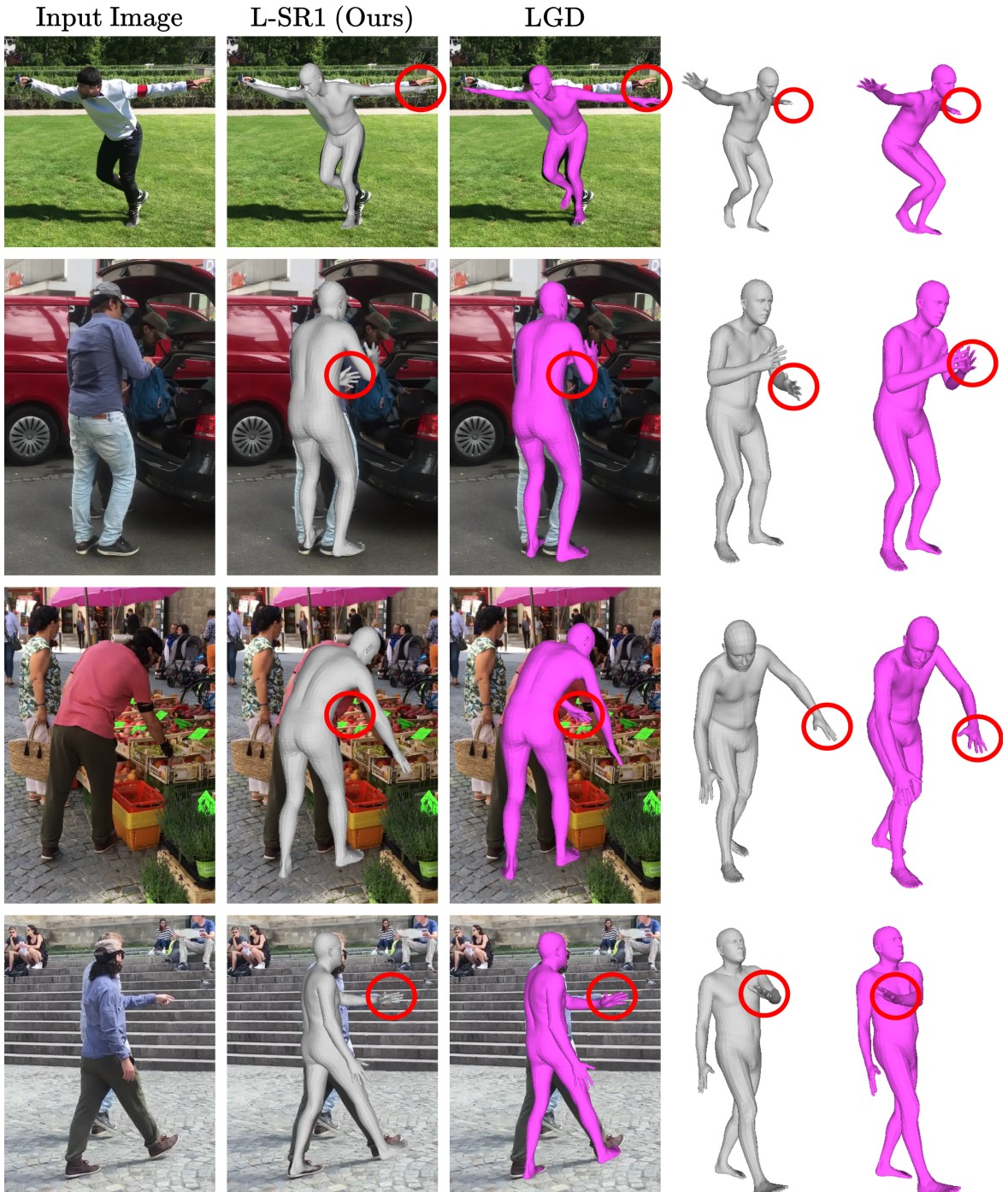

*Figure 12.* **Qualitative comparison of L-SR1 (ours) and LGD HMR results.** Meshes optimized with L-SR1 (Ours) are shown in white, while those from LGD (Song et al., 2020) are shown in pink. Regions of interest are highlighted with red circles.

# F. Theoretical Background

---

**Algorithm 2** Quasi-Newton (QN) Optimization

---

**Inputs:** Objective function $f \in \mathcal{C}^2$, initial point $\mathbf{x}_0 \in \mathbb{R}^n$

    Initialize $\mathbf{B}_0 \leftarrow \mathbf{I}$

    **for** $k = 1, 2, \ldots$ **do**

        $\mathbf{d}_k \leftarrow -\mathbf{B}_{k-1} \nabla f(\mathbf{x}_{k-1})$ {Compute descent direction}

        Choose $\alpha_k$ such that $f(\mathbf{x}_{k-1} + \alpha_k \mathbf{d}_k) < f(\mathbf{x}_{k-1})$ {Find step size}

        $\mathbf{x}_k \leftarrow \mathbf{x}_{k-1} + \alpha_k \mathbf{d}_k$ {Optimization step}

        $\mathbf{B}_k \leftarrow \text{UPDATE}(\mathbf{B}_{k-1}, \mathbf{x}_k - \mathbf{x}_{k-1}, \nabla f(\mathbf{x}_k) - \nabla f(\mathbf{x}_{k-1}))$ {Update $\mathbf{B}_k$}

    **end for**

**Output:** $\mathbf{x}^* \leftarrow \mathbf{x}_k$

---

**Algorithm 3** SR1 Update of Inverse Hessian Estimate

---

**Inputs:** Inverse Hessian estimate $\mathbf{B}_{k-1}$, step $\mathbf{p}_k \triangleq \mathbf{x}_k - \mathbf{x}_{k-1}$, gradient difference $\mathbf{q}_k \triangleq \mathbf{g}_k - \mathbf{g}_{k-1}$

    $\mathbf{v} \leftarrow \mathbf{p}_k - \mathbf{B}_{k-1} \mathbf{q}_k$

    **if** $\mathbf{v} \not\perp \mathbf{q}_k$ **then**

        $\mathbf{B}_k \leftarrow \mathbf{B}_{k-1} + \dfrac{\mathbf{v}\mathbf{v}^\top}{\mathbf{v}^\top \mathbf{q}_k}$ {SR1 update}

    **else**

        $\mathbf{B}_k \leftarrow \mathbf{B}_{k-1}$ {Skip update}

    **end if**

**Output:** $\mathbf{B}_k$

---

## F.1. Quasi-Newton Approach

Consider the Newton Method (NM), which utilizes the full Hessian matrix for preconditioning via directional adjustments. The NM minimization direction is defined as

$$\mathbf{d}_{\text{NM}} = -\mathbf{H}^{-1}(\mathbf{x}_k) \nabla f(\mathbf{x}_k), \tag{18}$$

where $\mathbf{H}^{-1}(\mathbf{x}_k)$ represents the inverse Hessian matrix evaluated at $\mathbf{x}_k$. Notably, when $f$ is quadratic, NM theoretically converges in a single iteration. However, the practicality of NM is often hindered by the challenge of computing and inverting the Hessian matrix. Consequently, a class of second-order optimization methods, termed Quasi-Newton (QN), emerged, aiming to approximate the inverse Hessian matrix, denoted $\mathbf{B}$, during the optimization process. This involves updating $\mathbf{B}$ iteratively alongside each optimization step. Algorithm 2 presents a summary of the QN optimization approach.

## F.2. Symmetric-Rank-One (SR1)

One prominent Quasi-Newton (QN) method is the Symmetric-Rank-One (SR1) technique, which involves iteratively accumulating symmetric rank-one matrices to estimate $\mathbf{B}$. Let $\mathbf{g}_k = \nabla f(\mathbf{x}_k)$ and $\mathbf{B}_k = \mathbf{H}^{-1}(\mathbf{x}_k)$ denote the gradient and inverse Hessian at step $k$, respectively. Considering the linear approximation of $\mathbf{g}_k$ as:

$$\mathbf{g}_k = \mathbf{g}_{k-1} + \mathbf{H}(\mathbf{x}_{k-1})(\mathbf{x}_k - \mathbf{x}_{k-1}), \tag{19}$$

and defining $\mathbf{p}_k = \mathbf{x}_k - \mathbf{x}_{k-1}$ and $\mathbf{q}_k = \mathbf{g}_k - \mathbf{g}_{k-1}$, equation (19) reduces to:

$$\mathbf{p}_k = \mathbf{B}_k \cdot \mathbf{q}_k. \tag{20}$$

This equation imposes a constraint directly on $\mathbf{B}_k$, known as the "secant constraint".

The SR1 method involves updating $\mathbf{B}$ with rank-one matrices of the form:

$$\mathbf{B}_k \leftarrow \mathbf{B}_{k-1} + \mathbf{u}\mathbf{v}^\top, \tag{21}$$

where $\mathbf{u}, \mathbf{v} \in \mathbb{R}^N$. Assuming $\mathbf{B}_{k-1}$ is symmetric, $\mathbf{B}_k$ is symmetric as well. Enforcing the secant constraint given in equation (20) on $\mathbf{B}_k$ yields:

$$\mathbf{p}_k = \left(\mathbf{B}_{k-1} + \mathbf{u}\mathbf{v}^\top\right) \cdot \mathbf{q}_k, \tag{22}$$

which can be rearranged as:

$$\mathbf{u} = \frac{\mathbf{p}_k - \mathbf{B}_{k-1}\mathbf{q}_k}{\mathbf{v}^\top \mathbf{q}_k}. \tag{23}$$

This implies that for every $\mathbf{v} \not\perp \mathbf{q}_k$, a suitable $\mathbf{u}$ satisfying the secant constraint can be found. A common choice is $\mathbf{v} = \mathbf{p}_k - \mathbf{B}_{k-1}\mathbf{q}_k$, leading to the following SR1 update step:

$$\mathbf{B}_k \leftarrow \mathbf{B}_{k-1} + \frac{\left(\mathbf{p}_k - \mathbf{B}_{k-1}\mathbf{q}_k\right)\left(\mathbf{p}_k - \mathbf{B}_{k-1}\mathbf{q}_k\right)^\top}{\left(\mathbf{p}_k - \mathbf{B}_{k-1}\mathbf{q}_k\right)^\top \mathbf{q}_k}. \tag{24}$$

The SR1 'Update B' process, summarized in Algorithm 3, notably ensures the symmetry of the estimated $\mathbf{B}$, a desirable feature. However, it falls short in guaranteeing positivity, a crucially property, as emphasized in the subsequent lemma.

**Lemma F.1.** *Let $f \in \mathcal{C}^2$ be an objective function, and $\mathbf{d}_{NM} = -\mathbf{B}\,\mathbf{g}$ represent the Newton direction. Then, $\mathbf{B} \succ \mathbf{0}$ ensures that $\mathbf{d}_{NM}$ is a descent direction.*

*Proof.* $\mathbf{d}_{\text{NM}}$ is a descent direction if and only if the directional derivative of $f$ in the direction $\mathbf{d}_{\text{NM}}$ satisfies $f'_{\mathbf{d}_{\text{NM}}} \leq 0$. Since $f'_{\mathbf{d}} = \mathbf{g}^\top \mathbf{d}$,

$$0 \geq f'_{\mathbf{d}_{\text{NM}}} = \mathbf{g}^\top \mathbf{d}_{\text{NM}} = -\mathbf{g}^\top \mathbf{B}\,\mathbf{g} \Leftrightarrow \mathbf{g}^\top \mathbf{B}\,\mathbf{g} \geq 0 \tag{25}$$

Hence, $\mathbf{B} \succ \mathbf{0}$ is a sufficient condition ensuring $\mathbf{d}_{\text{NM}}$ is a descent direction. $\square$

Consequently, several methods have been proposed to either eliminate 'bad' directions or ensure positive matrices through more sophisticated schemes.

