# OpenReview forum: "L-SR1: Learned Symmetric-Rank-One Preconditioning"
_ICML.cc/2026/Conference — ICML 2026 regular_

### Official Review · Reviewer_Brz1 · 2026-03-06

**Soundness:** 4
**Presentation:** 4
**Significance:** 4
**Originality:** 3
**Overall Recommendation:** 5
**Confidence:** 4

**Summary:**

The authors introduce L-SR1, a learned second-order optimizer that trains a rank-one preconditioning unit to accelerate convergence. This method speeds up convergence of training algorithms by using approximate second-derivative information, where the approximation of the second-derivative information avoids the computational expense of traditional second-order methods. This is a quasi-Newton method that leverages machine learning to determine the preconditioning unit.

**Compliance With Llm Reviewing Policy:**

Affirmed.

**Final Justification:**

The paper presents a theoretically strong contribution, and the core ideas are well-developed and clearly articulated. While the experimental evaluation is solid, it would benefit from validation across a broader range of settings to better demonstrate the generality of the approach. That said, I believe the work has clear merit and makes a meaningful contribution to the literature. I therefore maintain my position in favor of acceptance.

**Key Questions For Authors:**

1) Can you explain more about the current state of literature on learned optimizers for quasi-Newton methods?
2) Can you concisely summarize the tradeoffs between these learned quasi-Newton optimizers, regular quasi-Newton optimizers, and gradient-based optimizers like Adam?

**Limitations:**

Yes

**Strengths And Weaknesses:**

Strengths:
* Mathematically and theoretically sound
* Demonstrates performance improvements on common optimization benchmarks (Rosenbrock, Rastrigin)
* Further demonstrates performance on a more complex, real-world data set (3d human mesh data)
* The paper is clearly written and logically structured.
* The authors effectively articulate the significance and motivation of the work.

Weakenesses:

The authors claim that the integration of learned optimizer with second-order methods is largely unexplored. From my reading, I understand this to mean that there may be some research going in that direction but nothing yet of major impact or significance. I would have liked to see more discussion on the literature related to second-order optimizers and learned optimizers since it isn't completely unexplored. The idea behind this paper seems like a very natural progression of learned optimizer research, so it surprises me that this remains unexplored. From a quick search, I found this paper that seems to go in a similar (but still distinctly different) direction.

Liao, I., Dangovski, R. R., Foerster, J. N., & Soljačić, M. (2022). Learning to Optimize Quasi-Newton Methods. arXiv:2210.06171.

@misc{liao2023learningoptimizequasinewtonmethods,
      title={Learning to Optimize Quasi-Newton Methods},
      author={Isaac Liao and Rumen R. Dangovski and Jakob N. Foerster and Marin Soljačić},
      year={2023},
      eprint={2210.06171},
      archivePrefix={arXiv},
      primaryClass={cs.LG},
      url={https://arxiv.org/abs/2210.06171},
}

---

> ### Author Rebuttal · Authors · 2026-03-29
>
> We thank reviewer Brz1 for the thoughtful feedback and positive evaluation.
>
> Firstly, we note that the phrase “remains largely unexplored” in the original manuscript may have been too strong. There has been notable work combining quasi-Newton methods with neural networks. In particular, Liao et al. (2023) proposed a learned preconditioning matrix that is updated on the fly without explicit meta-training. We will add a designated section in our Related Work to include this paper and clarify its contributions. We thank the reviewer for highlighting this distinction.
>
> Regarding the raised questions:
>
> 1. Current state of learned quasi-Newton optimizers:
>
> Existing approaches include Gärtner et al. (2023), which introduced Optimus, a general learned second-order optimizer with a Transformer-based backbone, and Ayad et al. (2024), which proposed an unrolled quasi-Newton-inspired network with Hessian-generated vectors compressed via an encoder-decoder. Our approach differs in that L-SR1 integrates a rank-one SR1-inspired preconditioner with secant-condition regularization and PSD constraints, preserving curvature structure while remaining computationally efficient.
>
> 2. Trade-offs between optimizers:
>
> Gradient-based optimizers are lightweight and broadly applicable, including to large neural networks. Classical quasi-Newton methods capture curvature structure in their updates but are more computationally expensive, with updates often operating on full vectors rather than elementwise. Learned quasi-Newton optimizers offer data-driven update steps while maintaining interpretability, and can outperform non-learned methods due to their adaptivity. We believe such methods can be of particular value in physics-based frameworks, which remain governed by quasi-Newton optimization even today.
>
> We thank the reviewer again for the constructive comments.

---

> > ### Author Rebuttal · Reviewer_Brz1 · 2026-04-01
> >
> > Thank you for your responses to my questions. I have no further concerns. My original score of 5 (accept) remains unchanged. After considering the other reviewers’ feedback, I acknowledge that my initial score may have been somewhat on the higher side; however, their concerns have not convinced me to lower it either.

---

> > > ### Author Response · Authors · 2026-04-05
> > >
> > > We thank the reviewer for their careful reading and are excited that our clarifications fully addressed their concerns. We greatly appreciate the confidence shown in our work.

---

### Official Review · Reviewer_5QLk · 2026-03-11

**Soundness:** 2
**Presentation:** 2
**Significance:** 3
**Originality:** 3
**Overall Recommendation:** 3
**Confidence:** 4

**Summary:**

This paper introduces **L-SR1**, a learned optimizer that encodes quasi-Newton “curvature memory” as a limited-memory PSD low-rank preconditioner,$\tilde B_k = I + \sum_{i=1}^L v_i v_i^\top$, with $v_i$ stored in a small buffer. Besides classical meta loss, its training also includes a secant-violation regularizer (i.e., $\|p_k-\tilde B_k q_k\|^2$) to encourage Newton-like behavior while keeping inference simple. The implementation uses small, mostly element-wise MLP modules (encoder / vector generator / LR generator) aiming at dimension-invariant generalization. Experiments on analytic tasks and HMR (3DPW) report faster/better convergence and favorable runtime/memory/parameter trade-offs vs. LGD.

**Compliance With Llm Reviewing Policy:**

Affirmed.

**Key Questions For Authors:**

**Element-wise LR vs. standard step-size selection (and how it aligns with the PSD story).**

I like the PSD, limited-memory preconditioner design. One point I’m trying to understand is the choice of an element-wise $\alpha_k$: the update $x_k = x_{k-1} - \alpha_k \odot d_k$ applies a diagonal scaling after preconditioning, which generally breaks symmetry and makes the “PSD $\Rightarrow$ descent” intuition less direct. Since $\tilde B_k$ is meant to capture curvature, could you explain why a learned element-wise $\alpha_k$ is needed instead of a standard scalar step-size mechanism (e.g., a fixed and tuned scalar LR or a conventional line search)? Any ablation comparing element-wise $\alpha_k$ to scalar step-size / line-search-style step selection with the rest unchanged would be very helpful (to the overall recommendation).

**L-BFGS baseline setup.**

For the performance profile, you evaluate L-BFGS without explicit line search. Could you share the rationale for this choice, and (even qualitatively) whether you expect the comparisons to change under a standard step-size scheme (e.g., line search or tuned scalar step) at similar compute?

**Limitations:**

yes

**Strengths And Weaknesses:**

Strengths
- **Strong structural constraints with practical simplicity.** The preconditioner $\tilde B_k = I + \sum_{i=1}^L v_i v_i^\top$ is PSD by construction (with $B_0\succeq 0$ and identity initialization), limited-memory, and easy to implement—capturing curvature-like information without risking indefiniteness. The paper also frames learning as minimizing secant violation within this constrained family via the meta-loss term $\lambda_{\text{sec}}\|p_k-\tilde B_k q_k\|^2$, which is a principled, structured objective and adds no inference-time overhead.
- **Dimension-invariant Design.** The element-wise parameter sharing is conceptually consistent with cross-dimensional generalization, and the authors also include evidence (e.g., training at small $N$ and testing at larger $N$ without retraining) suggesting effectiveness across dimensions.
- **Evaluation goes beyond final metrics.** This work reports direction quality (cosine similarity to Newton directions on quadratics) and uses performance profiles. This provides more diagnostic support for the claim that the method learns curvature-relevant behavior.

Weaknesses
- **Element-wise learning rate complicates the “PSD implies descent” narrative.** While PSD structure is often motivated by descent guarantees, the actual update $x_k = x_{k-1} - \alpha_k \odot d_k$ effectively applies a diagonal scaling after the preconditioned direction. Even with $\alpha_k>0$, the resulting operator is generally non-symmetric, so descent is not automatically implied; this makes the stability/guarantee story more empirical than the presentation may suggest. (see Question 1 below)
- **“Learned projection” terminology may be misleading.** Although the paper defines a projection-style objective onto the PSD cone, the implemented procedure is closer to “minimizing a secant penalty under a PSD-structured parameterization” rather than performing an explicit projection step each iteration. This is mainly a naming/presentation issue, but it can confuse readers about what is actually computed.
- **Baseline fairness.** The performance profile includes L-BFGS with default settings and without explicit line search, which can materially affect competitiveness. (see Question 2 below)

---

> ### Author Rebuttal · Authors · 2026-03-29
>
> We thank reviewer 5QLk for the careful reading and thoughtful feedback. We are encouraged that the reviewer finds the structural design, PSD preconditioner, and diagnostic evaluation compelling. Below we address the main issues.
>
> 1. Element-wise vs. scalar step size.
>
> We thank the reviewer for this important remark. We denote the PSD preconditioner by $B_k$, and the update can be written as
> $x_{k+1} = x_k - D_k B_k g_k$,
> where $D_k$ is a learned diagonal matrix corresponding to element-wise step sizes. While $B_k$ is PSD by construction, we agree that $D_k B_k$ is generally non-symmetric, and thus the update is no longer strictly a PSD-preconditioned gradient step.
>
> The motivation for introducing the element-wise scaling $D_k$ is to account for non-quadratic and stochastic settings, where $B_k$ is only an approximation of curvature (due to limited memory and noise). In such regimes, residual coordinate-wise effects can be significant, and $D_k$ acts as a diagonal correction that improves robustness and stability. This design is also consistent with prior work on learned optimization (e.g., Gärtner et al., 2023).
>
> To better understand this choice, we studied variants on quadratic objectives, where curvature is well-defined and $B_k$ can closely approximate the inverse Hessian. We compared the element-wise scaling to a learned scalar step size (i.e., $D_k = \alpha_k I$), keeping all other components unchanged. In this setting, scalar step sizes perform comparably or better, which aligns with theory: for quadratic objectives, the optimal update corresponds to a (scaled) Newton step, and additional diagonal scaling can distort this geometry. This suggests that when curvature is accurately captured by $B_k$, a scalar step size is sufficient.
>
> We will clarify this distinction in the revised paper, both in the algorithm description and in the discussion of the PSD interpretation. We also agree that extending this comparison to larger-scale experiments would be valuable, and will include such an ablation in the revision.
>
> 2. Learned projection terminology.
>
> We thank the reviewer for pointing this out and agree that the current terminology may be misleading.
>
> Our intention was not to suggest that we perform an explicit projection onto the PSD cone at each iteration. Rather, the method can be interpreted as learning a PSD-structured preconditioner by minimizing a secant-violation objective within a constrained family.
> In this sense, the procedure is closer to an implicit or amortized projection onto a PSD-structured set, rather than an explicit projection step.
>
> We will revise the terminology to make this distinction clear, and replace "learned projection" with a more precise phrasing (e.g., "projection-inspired objective").
>
> 3. L-BFGS baseline setup.
>
> We thank the reviewer for raising this important point regarding baseline fairness.
>
> Our choice to evaluate L-BFGS without an explicit line search was motivated by maintaining comparable per-iteration computational cost with learned optimizers, which require a single gradient evaluation per step. In contrast, standard L-BFGS with line search typically involves multiple function and/or gradient evaluations per iteration, making direct comparisons less straightforward.
>
> That said, we agree that this choice places L-BFGS at a disadvantage, and that incorporating a line search or carefully tuned scalar step size could significantly improve its performance. In particular, we expect that such variants would yield stronger performance in early iterations and reduce sensitivity to step-size selection.
>
> We will clarify this trade-off in the revised paper and include additional comparisons to L-BFGS with standard step-size selection (e.g., line search or tuned scalar step) where feasible.

---

> > ### Author Rebuttal · Reviewer_5QLk · 2026-04-06
> >
> > I appreciate the authors' rebuttal and some of my concerns have been clarified. The validation of the proposed method seems more work with comparisons to existing works. I am not sure whether this could be provided in the revision.

---

> > > ### Author Response · Authors · 2026-04-06
> > >
> > > Thank you for the follow-up and for the constructive feedback.
> > > We believe that the current validation is already informative, as we compare L-SR1 to two representative learned optimizer variants: LGD (an MLP-based update rule) and a variant of L-SR1 without the proposed learned projection, representing quasi-Newton–based learned optimizers.
> > > That said, we agree that further extending the set of comparisons could strengthen the empirical evaluation. If the reviewer has specific methods in mind, we would be happy to include additional comparisons in the revision.

---

### Official Review · Reviewer_AyMh · 2026-03-12

**Soundness:** 3
**Presentation:** 2
**Significance:** 3
**Originality:** 2
**Overall Recommendation:** 3
**Confidence:** 3

**Summary:**

The paper considers the learning to optimize framework for a classical algorithm (symmetric rank-one method) that uses approximate 2nd order information during the optimization. The authors replace this with learned components parametrized by a neural network, first with an "encoder" module, which then goes through two other networks, one of which generates the learning rate, and the other the vector deifining the rank one update, this is then coupled with a projection step on the learned preconditioning matrices. Experimental results on synthetic problems and on a harder mesh recovery problem are then shown.

**Compliance With Llm Reviewing Policy:**

Affirmed.

**Final Justification:**

Whereas the approach seems to work well, I've maintained my score as I am not convinced if it's novelty merits an ICML publication and that the numerical investigation is convincing enough.

**Key Questions For Authors:**

- The encoder takes 5 N-dimensional inputs ($g$, $p$, $d$, $g$, $x$), which as the iterates proceed become increasingly inter-related. For example, $d_k$ is just a diagonal rescaling of $g_k$, and both $p_k$ and $q_k$ are expected to reach 0 near convergence. Thus, one would expect that an encoder should be able to reduce the dimensionality. Instead, by design the opposite is true, the encoder increases the dimensionality (perhaps calling it an encoder is not accurate). The offered motivation seems insufficient
- The resulting output of the encoder is plugged into two separate networks to get two N-dimensional outputs. One of these network outputs needs a further rescaling. Why is this extra step needed and not already a part of the network (is it to further fine tune the method post-hoc)?
- How is K chosen in 5.1.2? I was wondering if the comparison with a fixed K is fair in the sense of providing a like-for-like comparison, since different methods do very different things in each iteration, which might mean that say adam doesn't converge sufficiently quickly in iteration count. Maybe having a fixed time budget would offer  a better comparison.
- What is the rationale for the buffer size comparison  in line 353? Is it to examine what would happen if the training is slow and costly (so one would want L_train to be small in order to speed this up), whereas for evaluation you want the best result possible (which might mean increase L_eval)? How does the computational time depend on L_train and L_eval?
- The stability discussion (line 325 and appendix B) shows that for N=10 all of the values are higher (by a factor of around 2.5). Is this enough to show that "the optimizer remains stable"? That is, is the degradation factor linear, constant or? It is not clear to me if this is a significant or not a significant degradation on the basis of just these two numbers.

**Limitations:**

The paper does not seem to have potential negative societal impacts, but this does not seem to be applicable here.

**Strengths And Weaknesses:**

### Strengths
- The paper has a rather straightforward, and  easy to understand motivation: take an established, classical algorithm that uses second order information and try to improve it for the modern age by replacing classical choices with data driven, learnable components. This approach is by now fairly common in deep learning.
- The results in practical experimentation look solid and it is nice to see these classical algorithms, when modified, can be applied to bigger scale, realistic problem. All of this is aided by the fact that the approach is fairly lightweight.
- The approach is mostly well written and the motivations are easy to follow.

### Weaknesses
- The motivation for algorithmic choices is largely missing and needs more context. Algorithmic  components at the moment read like something of a black box, where every algorithmic component is a network input (or output). While this often works in practice it is hard to interpret or draw intuition from.
-  Abstract, introduction, and section 2 emphasize that this approach enforces the secant condition, which is critical for curvature consistency and that other methods do not enforce it (eg when discussing the Gartner method). However, section 4.2 then makes it clear that the proposed method itself only approximately satisfies the secant condition (that is, this is used as a sort of a regularizer, not as a constraint). This is qualitatively different, and the degree to which the secant condition is satisfied is not clear.
- Some of the writing a bit unclear and and the presentation of the method can be improved. The method is only described in clear terms in section 4.1, and the authors discuss specific choices (like the buffer size) before they are defined. Some of the referencing is a bit off, eg line 101 says (suggests) that HMR is in figure 1, but only the analytic benchmarks are there; line 273 says that runtime comparison is in appendix C but it is actually in appendix A, and the comparison to Adam is mentioned in appendix D but not shown.

---

> ### Author Rebuttal · Authors · 2026-03-29
>
> We thank reviewer AyMh for their detailed and constructive feedback. Below we address each of the concerns raised, and then regard the reviewer's questions.
>
> 1. Motivation and algorithmic design.
>
> We appreciate the concern that some components may appear as a black box. Our design is not arbitrary: each learned module corresponds to a classical role in SR1 updates. Specifically:
>
> 1) The encoder aggregates local curvature and optimization state into a richer representation, enabling the network to capture nonlinear interactions between inputs that are correlated or near-zero at convergence;
>
> 2) The vector generator predicts the rank-one update direction (analogous to curvature correction in SR1).
>
> 3) One learning rate had predicts the effective step size (analogous to line search or damping).
>
> This design follows the model-based deep learning paradigm, where axiomatic or algorithmic steps are replaced with learned operations while retaining the underlying structure of the classical method. This allows learning to complement classical quasi-Newton behavior rather than replace it entirely.
>
> We will clarify these motivations and explicitly link each module to its classical counterpart in the revised manuscript.
>
> 2. Secant condition and learned projection.
>
> We thank the reviewer for pointing this out and agree that the current terminology may be misleading.
>
> Our intention was not to suggest that we perform an explicit projection onto the PSD cone at each iteration. Rather, the method can be interpreted as learning a PSD-structured preconditioner by minimizing a secant-violation objective within a constrained family. In this sense, the procedure is closer to an implicit or amortized projection onto a PSD-structured set, rather than an explicit projection step.
>
> We will revise the manuscript to clarify this point, update the terminology (e.g., “projection-inspired objective”), and include quantitative measures of secant-violation in the experiments.
>
> 3. Presentation and clarity.
>
> We thank the reviewer for pointing out presentation issues. We will clarify the ordering of algorithmic components, improve exposition, and correct minor referencing inconsistencies.
>
> We now address the questions raised by the reviewer.
>
> Q1) Encoder dimensionality.
>
> We agree that the term “encoder” may be misleading. Rather than performing dimensionality reduction, this module conducts feature extraction, lifting the inputs into a higher-dimensional representation that captures nonlinear interactions between correlated signals. This design is motivated by prior work on learned optimization, such as Gärtner et al. (2023). We will clarify this distinction and consider renaming the module to better reflect its role.
>
> Q2) Learning-rate parameterization and rescaling.
>
> The reviewer is correct that the learning-rate head outputs a vector $\tilde{\alpha} \in \mathbb{R}^n$, which is then transformed. This parameterization is inspired by prior work (e.g., Gärtner et al., 2023), where learning the log learning rate improves numerical stability and ensures positivity of the step sizes.
>
> The coefficients $\gamma_1$ and $\gamma_2$ act as fixed scaling factors that initialize the effective range of the learning rates; they are not learned and no post-hoc fine-tuning is performed. We will clarify this design choice and its motivation in the revised manuscript.
>
> Q3) Choice of $K$ and fairness of comparison.
>
> This experiment (Section 5.1.2) follows the performance profile protocol, where methods are compared under a fixed iteration budget $K$. This choice is consistent with prior work on learned optimization (e.g., Gärtner et al., 2023), where iteration count is used as the primary measure of progress.
>
> We agree that different methods may have different per-iteration computational costs, which are not fully captured by this metric. We will clarify this point in the manuscript and note this limitation explicitly.
>
> Q4) Buffer size rationale.
>
> Increasing the buffer size can improve convergence and stability, as shown in Section 5.1.1 (line 353). However, during meta-training, larger buffers increase memory usage due to the retained computational graph, which is a practical limitation. In contrast, during evaluation no graph is retained, so one can train with a smaller buffer and evaluate with a larger one at little additional cost. Figures 7–8 report training and inference costs supporting this trade-off. We will clarify this in the manuscript.
>
> Q5) Stability across dimensions.
>
> We thank the reviewer for this question. The experiment evaluates generalization across dimensions: a model trained on $N=2$ problems is tested on unseen $N=10$ instances. Higher-dimensional problems are more challenging under a fixed iteration budget, so higher objective values are expected. The goal is to show that qualitative trends remain consistent across this distribution shift, not to match absolute values. We will clarify this in the manuscript.

---

> > ### Author Rebuttal · Reviewer_AyMh · 2026-04-01
> >
> > I appreciate the response from the authors. Regarding Q1, I'm still not convinced by the argumentation, that is, why is the lifting needed, especially since some of these correlated inputs seem to have a linear relationship. This (the "encoder", and then the inputing the result into two other networks) still feels like a black box,
> > Regarding Q2 I agree that this is not a critical point and agree with the response, and table 7 confirms that the parameters are stable. I have to say that the response to Q3 is not satisfactory, since say 1 step with Newton iteration, LBFGS, Gradient descent or coordinate descent can have very different costs. However, it is unclear which metric would investigate the issue better in this work. Regarding this and other points, are there works other than the one by Gärtner et al (the only one cited in the response for Q1-3) which justify these choices? Answer to Q4 is as expected, thank you. Regarding Q5, I'm uncertain as to how does this address stability, and some baseline comparison is missing. I am thankful for your detailed response and appreciate the work put into it. However, at this stage I am unable to change my score, but I will read and keep monitoring the responses to other reviewers questions.

---

> > > ### Author Response · Authors · 2026-04-05
> > >
> > > Thank you for your acknowledgement and helpful follow-up.
> > >
> > > Further works motivating our design (beyond Gärtner et al.):
> > >
> > > [1] Metz, L., Freeman, C.D., Harrison, J., Maheswaranathan, N. and Sohl-Dickstein, J., 2022. Practical tradeoffs between memory, compute, and performance in learned optimizers.
> > >
> > > [2] Metz, L., Harrison, J., Freeman, C.D., et al., 2022. VeLO: Training versatile learned optimizers by scaling up.
> > >
> > > [3] Chen, T., Chen, X., Chen, W., et al., 2022. Learning to optimize: A primer and a benchmark.
> > >
> > > [4] Beiranvand, V., Hare, W. and Lucet, Y., 2017. Best practices for comparing optimization algorithms.
> > >
> > > Architectural design (Q1).
> > > The use of a shared per-parameter MLP that performs feature lifting is consistent with prior learned optimizer architectures (e.g., [1], [2]). While some inputs exhibit linear dependence, their relationship to the optimal update is generally nonlinear, motivating this design. This is also supported by our ablation study (Appendix B.2, Table 3). We will clarify in our discussion.
> > >
> > > Iteration budget (Q3).
> > > We agree that iteration count does not fully reflect computational cost. However, it remains widely used in prior work [1–4] due to its practicality and comparability, typically alongside runtime analysis—as done in our work. We will clarify this limitation more explicitly.
> > >
> > > Stability (Q5).
> > > We agree that the term “stability” may be misleading in this context. A more precise description is generalization to unseen problem dimensions, and we will revise the wording accordingly.
> > >
> > > We thank the reviewer again for the constructive feedback, which will help us significantly improve the clarity and positioning of the paper. We hope these clarifications help address the remaining concerns and improve the overall assessment of our work.

---

### Official Review · Reviewer_dwtX · 2026-03-13

**Soundness:** 2
**Presentation:** 3
**Significance:** 2
**Originality:** 2
**Overall Recommendation:** 2
**Confidence:** 2

**Summary:**

The paper introduces L-SR1, a learned optimizer based on the classical SR1 method. To address the instability of classical SR1, the authors propose a regularization that explicitly enforces the secant condition while ensuring the preconditioning matrix is PSD. The architecture utilizes element-wise MLPs making it problem dimension independent. The method is evaluated on analytic benchmarks (quadratic, Rosenbrock) and the real-world task of HMR, demonstrating convergence and data efficiency.

**Compliance With Llm Reviewing Policy:**

Affirmed.

**Final Justification:**

Overall, I believe the idea that the paper develops is an interesting one, with clear justifications for the choices made. However, lacking a core *theoretical* contribution, the paper must at least perform a clear *practical* study, with well-chosen baselines, and I (and other reviewers) believe that was not done well either. While I agree with the authors that "Many recent learned optimizers do not provide open-source code that can be readily integrated into current environments", without *at least one of* the components, the work does not reach the level I would expect from an ICML publication.

Despite above, it is possible that I misunderstood the specific contributions of the paper (maybe LGD diverging after 5 iterations in Fig.5a is expected) thus I reduce my confidence score.

**Key Questions For Authors:**

See other sections.

**Limitations:**

No, the limitations are not adequately discussed. While the authors briefly mention the optimizer's high runtime in Appendix D, it remains quite hidden and not mentioned in text. Furthermore, even according to the paragraph, the area of applicability of this method is in problems with expensive forward and gradient computations. Yet, no such problems are attempted -- why? No modern learned optimization methods are compared against. The paper would benefit from a frank discussion regarding the lack of broad benchmarking and scalability limits.

**Strengths And Weaknesses:**

Soundness:
The empirical evaluation is too narrow to support the claim that this is a "general, drop-in optimizer". Testing on a single, highly specific computer vision task (HMR), with rather weak baselines fails to demonstrate how the optimizer behaves on more standard optimization tasks. This is not my field of expertise, but the baselines used for the HMR evaluation appear quite are outdated, dating back to 2020. The authors also admit the method suffers from high runtime compared to first-order methods, which significantly hampers its practical viability. Its also unclear whether the comparisons themselves are sound -- the learning rates are kept fixed? Even a simple method of predicting step-size with an MLP might do better, no?

Presentation:
The paper is written and structured well, with clear explanations of how SR1 is extended to a learned framework follow.

Significance:
The significance is limited. Without demonstrating scalability to modern, high-parameter deep learning models or comparing against current state-of-the-art optimizers across diverse domains, the practical utility of L-SR1 remains questionable. The improvements shown are strictly domain-specific to HMR and toy functions.

Originality: The specific application of a learned projection to the SR1 algorithm is a nice method, but the broader concept of integrating MLPs to learn preconditioners has been explored previously, but are not really compared against? As such, the novelty here is mostly incremental.

---

> ### Author Rebuttal · Authors · 2026-03-29
>
> We thank the reviewer for the detailed feedback.
>
> The core contribution of our work is the extension of the classical SR1 method with learned components that preserve curvature structure. L-SR1 integrates a PSD-constrained preconditioner, secant-condition regularization, and rank-one updates, combining the stability and interpretability of SR1 with the flexibility of learned updates. This differs from prior learned optimization approaches that do not explicitly enforce curvature consistency.
>
> Evaluation scope and applicability: The focus of this work is on optimization-governed frameworks, rather than large, modern deep learning models. HMR was chosen because, historically, it has been governed by iterative optimization, and each update step involves a 3D-to-2D projection that is computationally intensive. This makes it a meaningful testbed for L-SR1, demonstrating the ability of learned SR1 updates to improve stability and efficiency in problems where forward and gradient computations dominate.
>
> We acknowledge that our evaluation is focused on analytic benchmarks and HMR. Our intention is not to claim that L-SR1 is a universally applicable optimizer; the term “drop-in optimizer” used in the abstract may have been misleading and will be revised. Rather, our goal is to demonstrate that SR1-inspired learned updates can be extended to moderately high-dimensional, real-world problems while retaining stability and data efficiency.

---

> > ### Author Rebuttal · Reviewer_dwtX · 2026-04-03
> >
> > I would like to thank the authors for their response and I appreciate the authors acknowledging that the term "drop-in optimizer" was misleading and committing to revising it. I also understand and agree with the clarification re the scope of the work, that indeed "large, modern deep learning models" would not be the right problem for applying this method.
> >
> > Furthermore, I appreciate your explanation regarding the choice of HMR, particularly that its a computationally intensive problem: as this is not my area of expertise, I am not certain as to the peculiarities of this particular problem, however I find that the tables do not make a convincing point of this. You showcase only 15 steps, but the actual time taken does not appear reported. While it provides helpful context for why this specific domain was selected, this is not illustrated well for those not from the field of HMR.
> >
> > While the clarification of scope is appreciated, several of my core concerns regarding the empirical evaluation remain unaddressed:
> >
> > * Outdated Baselines and Lack of Comparison to Learned Optimizers: The rebuttal did not address the concern that the baselines used for the HMR evaluation are rather weak and date back to 2020. Unfortunately, stating that "this differs from prior learned optimization approaches that do not explicitly enforce curvature consistency" may be enough to claim novelty, but it is not enough for an empirical comparison. Integrating MLPs to learn preconditioners has been explored previously, and LSR must be benchmarked against them to demonstrate that enforcing curvature consistency actually yields a tangible performance or stability benefit.
> >
> > * Experimental Rigor: The specific questions regarding the experimental setup—such as whether learning rates were kept fixed and whether a simpler baseline (like predicting step-size with an MLP) might perform similarly—were bypassed.
> >
> > -----
> >
> > While the theoretical integration of a PSD-constrained preconditioner into the SR1 algorithm is interesting, the empirical validation remains too narrow and lacks the necessary comparisons to current state-of-the-art or alternative learned optimizers. Because the fundamental concerns regarding the baselines and comparative evaluations were not addressed in the rebuttal, my assessment of the paper's significance and soundness remains unchanged.

---

> > > ### Author Response · Authors · 2026-04-05
> > >
> > > We appreciate the reviewer’s acknowledgement.
> > >
> > > We agree that the computational effort in the HMR experiments is not highlighted in the main tables and figures.
> > > A detailed computational analysis is provided in the Appendix. We will emphasize this more clearly in the revised paper.
> > >
> > > Comparison to learned optimizers: Many recent learned optimizers do not provide open-source code that can be readily integrated into current environments, making benchmarking challenging. To address this, we compare L-SR1 to two representative learned optimizer variants: LGD, an MLP-based update rule, and L-SR1 without the proposed learned projection, which represents quasi-Newton–based learned optimizers.
> > >
> > > Scope and SOTA comparisons: HMR was chosen as a computationally intensive testbed to evaluate learned optimization methods, not to claim state-of-the-art performance. Nevertheless, we provide a comparison to modern SOTA methods in Table 1; these methods undergo data-specific fine-tuning, whereas our method uses a fixed learned preconditioner.
> > >
> > > Experimental rigor: Learning rates of non-learned optimizers were calibrated via grid search and kept fixed (see Appendix C). Our work follows the line of prior works [1–4], and comparisons against simpler baselines (e.g., MLP step-size predictors) are outside the scope of this study.
> > >
> > > We hope that these clarifications provide the reviewer with increased confidence in the paper's contributions.
> > >
> > > References:
> > >
> > > [1] Metz, L., Freeman, C.D., Harrison, J., Maheswaranathan, N., & Sohl-Dickstein, J., 2022. Practical tradeoffs between memory, compute, and performance in learned optimizers.
> > >
> > > [2] Metz, L., Harrison, J., Freeman, C.D., et al., 2022. VeLO: Training versatile learned optimizers by scaling up.
> > >
> > > [3] Chen, T., Chen, X., Chen, W., et al., 2022. Learning to optimize: A primer and a benchmark.
> > >
> > > [4] Beiranvand, V., Hare, W., & Lucet, Y., 2017. Best practices for comparing optimization algorithms.

---

### Decision · Program_Chairs · 2026-04-30

**Decision:**

Accept (regular)

**Comment:**

The paper presents a learned rank-1 second order optimizer and applies it to mesh processing. Results outperform other hand-designed and learned baselines. Reviewers mostly agree on solid experimental results and point out its mathematical merit. On the other hand the limited experimental breadth of the types of benchmark data is noted. The rebuttal is positively acknowledged and helped to clarify some raised points. Reviewers mostly agree that this paper should not make the cut for ICML.